



# Merging bio-optical data from Biogeochemical-Argo floats and models in marine biogeochemistry

Elena Terzić[1,5], Paolo Lazzari[1], Emanuele Organelli[2], Cosimo Solidoro[1], Stefano Salon[1], Fabrizio D'Ortenzio[3], and Pascal Conan[4]

[1]Istituto Nazionale di Oceanografia e di Geofisica Sperimentale - OGS, Via Beirut 4, 34151 Trieste, Italy
[2]Plymouth Marine Laboratory, Prospect Place, The Hoe, Plymouth PL1 3DH, United Kingdom
[3]Sorbonne Universités, UPMC Université Paris 06, CNRS, LOV, Villefranche-sur-Mer, 06230, France
[4]Sorbonne Université, Pierre et Marie Curie-Paris 06, CNRS - UMR7621 LOMIC, F66650 Banyuls-sur-Mer, France
[5]Università degli Studi di Trieste, Dipartimento di Matematica e Geoscienze, Via E. Weiss 2, 34128 Trieste, Italy

*Correspondence to:* Paolo Lazzari (plazzari@inogs.it)

**Abstract.** The present work is based on a dataset comprised of 31 Biogeochemical (BGC) Argo floats that collected 0-1000 m vertical profiles of biogeochemical and optical data from 2012 to 2016 in the Mediterranean Sea. The dataset was integrated

in 1-dimensional model simulations following the trajectories of each float and considering measured photosynthetically available radiation (PAR) profiles as the reference light parameterization. The simulations were aimed to be consistent with data measured by float sensors, especially in terms of the deep chlorophyll maximum (DCM) depth. Moreover, we tested several light models in order to estimate their impact on modeled biogeochemical properties, including self-shading dynamics based on chlorophyll and colored dissolved organic matter (CDOM) concentrations. The results, evaluated with the corresponding

in-situ BGC-Argo chlorophyll data, indicate that the proposed approach allows to properly simulate the chlorophyll dynamics and illustrate how PAR and vertical mixing are essential environmental regulation factors driving primary producers dynamics. The higher skills are reached using in-situ PAR, but some of the alternative bio-optical models here presented show comparable skill in reproducing DCM depth spatial variability. Simulation results show that during the stratification phase the diel cycle has significant impact on the surface chlorophyll regimes. The approach here presented serves as a computationally smooth

solution to analyse BGC-Argo floats data and to corroborate hypotheses on their spatio-temporal variability.

## 1   Introduction

The availability of radiometric profiles in the open ocean on a global scale has drastically increased due to an enhanced deployment of autonomous Argo floats with additional biogeochemical and optical sensors, officially termed as Biogeochemical-Argo floats (hereafter BGC-Argo floats; Johnson and Claustre, 2016). Their wide use has been also a consequence of refined sensor-

calibration and data quality-control procedures that have been applied on acquired profiles (see Organelli et al., 2016, 2017).



The potential of such observational tools might be further expanded with the upcoming introduction of satellite multi-band sensors (such as the OLCI sensor on board of the ESA Sentinel-3 mission[1] and the forthcoming NASA PACE program[2]), as well as in the development, calibration and tuning of bio-optical numerical models in the ocean (Dutkiewicz et al., 2015; Baird et al., 2016; Gregg and Rousseaux, 2017), that can in turn allow a more comprehensive investigation of the link between
physical and biogeochemical processes in the oceans.

Due to a high density of BGC-Argo floats and a generally low cloud sky coverage, the Mediterranean Sea represents an ideal location to carry out numerical experiments of that kind. The Mediterranean has clearly been identified as a "hotspot" for climate change, and is therefore expected to experience environmental impacts (de Madron et al., 2011). In addition, the Mediterranean Sea is characterized by complex trophic gradients (Crise et al., 1999; Lazzari et al., 2012; d'Ortenzio
and Ribera d'Alcalà, 2009) and spatially heterogeneous inherent and apparent optical properties (Oubelkheir et al., 2005). Such gradients are mainly related to the inverse estuarine circulation of the area (Crispi et al., 2001) and to the varying distribution of optically significant substances (e.g. colored dissolved organic matter - CDOM; non-algal particles - NAP) that modulate the light penetration along the water column (Morel and Gentili, 2009b). Moreover, inherent optical properties (IOPs) could be affected also by important processes of Saharan dust deposition (Claustre et al., 2002). Furthermore, the
Mediterranean Sea Monitoring and Forecasting Centre (Med-MFC) operatively produces analyses, forecasts and reanalyses of a series of biogeochemical state variables (e.g. chlorophyll, nutrients, pCO2) for the European Copernicus Marine Environment Monitoring Services (CMEMS) since 2015 using the MedBFM model (Lazzari et al., 2010, 2012, 2016), which assimilates surface chlorophyll from satellite observations (Teruzzi et al., 2014). However, it is important to explore the feasibility of the direct assimilation of radiometric quantities that appears more robust than the chlorophyll-based one as a result of a more
accurate uncertainty characterization for optical measurements (Dowd et al., 2014; Organelli et al., 2016) compared to other properties, such as fluorescence-derived chlorophyll *a* concentration. Specific studies are required to demonstrate to what extent the assimilation of radiometric data can improve the model skill in simulating key biogeochemical variables (e.g. nutrients, primary productivity).

In this paper we propose a methodology where radiometric quantities measured by BGC-Argo floats are embedded within a
1-dimensional (1D) numerical model in order to replicate the biogeochemical evolution of the water column observed by floats. For each float a separate simulation is carried out and the measured chlorophyll concentrations (i.e. derived from fluorescence) are compared with the simulated values.

Given the substantial number of profiles and their high vertical resolution, such simulations can be considered as a convenient evaluation tool of the chlorophyll spatio-temporal patterns along the water column by comparing them with the corresponding
in-situ measurements. Furthermore, the present method allows to implement various bio-optical models in order to estimate how well they perform compared to in-situ measurements of the photosynthetically active radiation (PAR). The objective of the present study is twofold: 1) to show how it is possible to integrate BGC-Argo float bio-optical data and a simple 1-D model to investigate chlorophyll vertical dynamics; 2) to use such a tool on a sufficiently large data set in order to test different bio-

---

[1]https://sentinel.esa.int/web/sentinel/user-guides/sentinel-3-olci

[2]https://pace.gsfc.nasa.gov/





optical models. The paper is organized as follows: in the Methods section, the Mediterranean Sea BGC-Argo floats network and the model configurations are presented. In the Results and Discussion section, we analyse the 1D biogeochemical simulations and their sensitivity according to the objectives of the work. General remarks are illustrated in the Conclusions section.

## 2  Methods

### 2.1  BGC-Argo floats data

The Mediterranean Sea BGC-Argo network operating in the period 2012-2016 was composed of 31 floats. The BGC-Argo floats acquired 1314 vertical profiles of temperature (T) and salinity (S), chlorophyll $a$ concentration (Chl $a$, units of $mg\,m^{-3}$), derived from fluorescence measurements between 0 and 1000 m (see Organelli et al., 2017; Roesler et al., 2017), and radiometric quantities, such as downward planar irradiance ($E_d$) at three different wavelengths ($\lambda$ = 380, 412 and 490 nm, units of $\mu W\,cm^{-2}\,nm^{-1}$ ) and Photosynthetically Active Radiation (PAR, unit of $\mu mol\,quanta\,m^{-2}\,s^{-1}$), which gives information on the penetration of light for the whole visible band (from 400 to 700 nm; Kirk, 1994). Radiometric measurements were obtained in the upper 250 m, with vertical resolution of 1 m between 10 and 250 m and 0.20 m between 0 and 10 m. All profiles were acquired around local noon. The quality control (QC) procedure for irradiance profiles consisted of dark signal and cloud identification, wave focusing and spikes correction (for a more detailed explanation see Organelli et al., 2016). Chlorophyll concentration QC was performed according to the procedure of the international BGC-Argo program (Schmechtig et al., 2016; Organelli et al., 2017). The 7 variables (T, S, Chl, $E_{d380}$, $E_{d412}$, $E_{d490}$, PAR) were then vertically interpolated to a resolution of 1 m in the upper 400 m. Note that the operational definition of PAR used by the BGC-Argo community takes into consideration the planar irradiance $E_d$ rather than the scalar one $E_o$, therefore differing from its theoretical definition and leading to an underestimation of its values by 30% or more (Mobley et al., 2010). The scalar values of PAR were thus derived according to Baird et al. (2016), although the correction related to the irradiance scattering was neglected due to the lack of information on IOPs. The second approach is based on a constant correction factor: in-situ experiments carried out in the Tyrrhenian Sea indicated that a correction factor of 1.58 can be applied to retrieve scalar from planar irradiance (Dr. Luca Massi, University of Florence, pers. comm.). The two approaches give consistent results with slightly higher skill in the first case, which was adopted in the experiments shown in the following sections.

BGC-Argo float sensors measure Chl $a$ concentration through its fluorescent property (Holm-Hansen et al., 1965) of absorbing blue light and re-emitting it at the red part of the spectrum (with the excitation at 470 and emission at 690 nm).

The ratio of fluorescence to Chl $a$ concentration is highly variable and it depends on the taxonomic composition of the algal species, environmental conditions (temperature and nutrient concentration; Kiefer, 1973), as well on physiological responses to light, such as photoacclimation (MacIntyre et al., 2002; Moore et al., 2006; Falkowski and LaRoche, 1991) and non-photochemical quenching (Cullen and Lewis, 1995; Falkowski and Kolber, 1995; Xing et al., 2011; Falkowski and Raven, 2013).

Due to a factory calibration bias in the Chl $a$ estimation from fluorescence, a correction factor of 0.5 was applied over the global ocean database for WETLABS ECO series Chl fluorometers after having compared fluorescence-derived values with



the ones obtained by high performance liquid chromatography - HPLC (Roesler et al., 2017; Barbieux et al., 2018; Organelli et al., 2017).

A simple geographic partition of profiles was performed with a spatial division into 13 (out of 16) sub-basins (Fig.1), with the majority of profiles located in the North Western Mediterranean (NWM, 332 profiles), followed by Northern Ionian (ION3, 172 profiles) and Southern Tyrrhenian (TYR2, 162 profiles). No data were available for the South-western Ionian (ION1) and the Eastern Levantine (LEV4) and only one profile was present in the Northern Adriatic (ADR1), as well as in the Western Levantine (LEV1). The WMO code specification for each BGC Argo float is provided in the supplementary material.

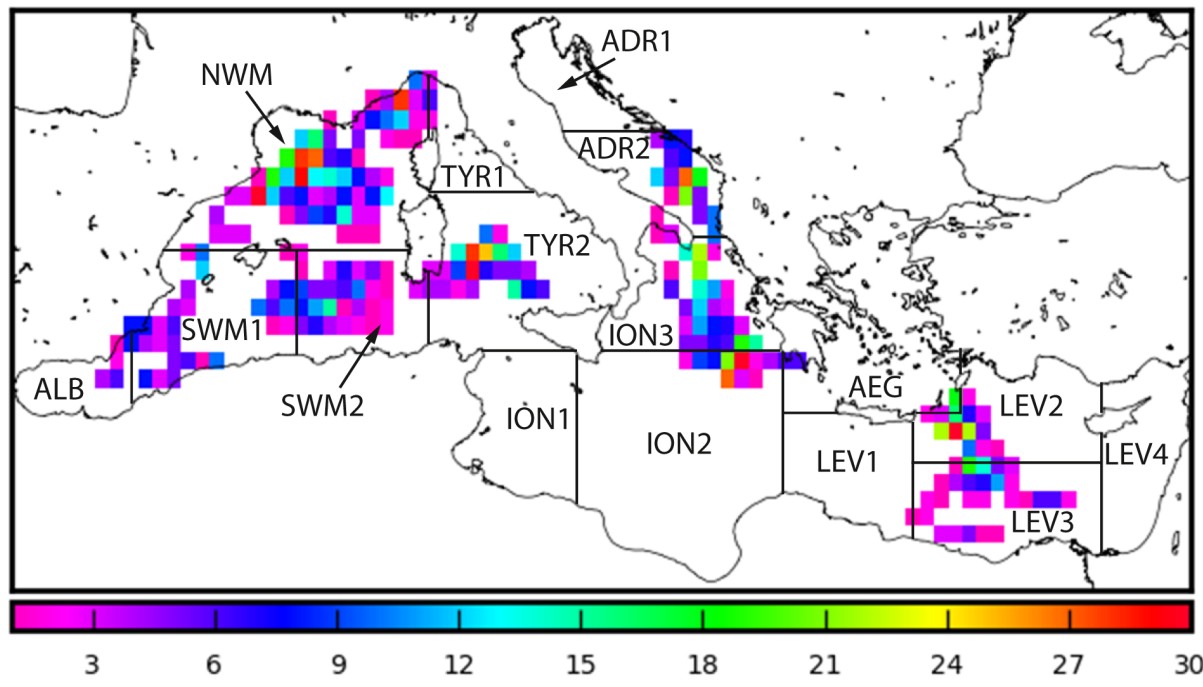

**Figure 1.** Spatial distribution of float profiles superimposed to sub-basin division used in the Mediterranean CMEMS system.

## 2.2 1-D Biogeochemical model

Biogeochemical processes have been simulated according to the voxel approach ("volume element with biological content and processes"; Kohlmeier and Ebenhöh, 2009), discretized along the vertical in order to resolve vertical irradiance attenuation and nutrient gradients. Each voxel replicated light and mixing conditions according to the trajectory and measurements of the corresponding BGC-Argo float, therefore simulating a pseudo-lagrangian experiment. No exchanges of mass between the voxel and the surrounding field have been considered, therefore implying that mass exchanges due to horizontal diffusion and baroclininc components of the (upper ocean) advection field are assumed to be smaller compared to vertical processes and biogeochemical dynamics. Conversely, the voxel exchanges heat with the atmosphere and receives light in accordance with its moving position. This approach, similar to the one already adopted by Kohlmeier and Ebenhöh (2009), has been already




successfully applied by Mignot et al. (2018) to analyze BGC-Argo Floats in the North Atlantic. Furthermore, we assumed that major biogeochemical transformations can be described by the Biogeochemical Flux Model parameterizations (see below), properly driven by a bio-optical model. These assumptions have been validated by contrasting model results and experimental data, as shown later.

In the first set of simulations the biogeochemical model was forced by PAR measurements obtained from BGC Argo floats. Experimental values of temperature and density (computed from float temperature and salinity profiles) are also taken into consideration. A simulation for each of the BGC-Argo float trajectories has been performed with this setup, hereafter abbreviated as REF. Four additional sets of simulations have been performed by using the same setup while considering four different values of vertical eddy diffusivity coefficients (MLD1, MLD2, MLD3 and MLD4) in order to assess uncertainties in REF

simulation due to uncertainties in the vertical diffusion parameterization. After having evaluated the REF model capability, we performed six additional sets of simulations by forcing the biogeochemical model with PAR obtained by alternative bio-optical models (OPT1, OPT2a,b,c,d). Moreover, we consider the impact of the bio-optical model approach currently used in the CMEMS Copernicus system (OPT3). In this way we assessed the possibility of using biogeochemical models also when PAR measurements are not available, and compared the skill of different bio-optical models. The final set of simulations has

been devoted to explore specific questions, such as the impact of using a constant light approximation rather than following the diurnal light variation (setups CL1 and CL2). Finally, a first attempt to model the impact of CDOM has been made by adding a new set of variables to both biogeochemical and bio-optical models (OPT4a,b,c,d and OPT5). The whole ensemble of simulations is summarized in Tab.1 and Tab.2.

The biogeochemical model BFM (Vichi et al., 2013) is a biomass-based numerical model that simulates the biogeochemical

fluxes of carbon, phosphorus, nitrogen, silicon, and oxygen, characterizing the lower trophic level (producers, consumers, and recyclers) of the marine ecosystem. Our application is based on the coupled transport-biogeochemical model OGSTM-BFM (Lazzari et al., 2012, 2016). The model includes four phytoplankton functional types (diatoms, nanoflagellates, picophyto-plankton, and dinoflagellates), carnivorous and omnivorous mesozooplankton, bacteria, heterotrophic nanoflagellates, and microzooplankton. Each variable is described in terms of internal carbon, phosphorus and nitrogen concentrations. Phytoplankton

functional types can be characterized regarding prognostic Chl and can additionally consider the silicate component in the case of diatoms. Particulate and dissolved organic matter are also included, the latter partitioned in the labile, semi-labile and semi-refractory phases. The full BFM parameters specification is provided in the supplementary material. Here we focus mainly on total Chl a, reserving to future analysis (according to data availability and optical model complexity) a study of the Plankton Functional Types (PFT) resource competition dynamics and other important aspects of the marine ecosystem.

In this application, the vertical resolution is 1 meter. Initial conditions for all biogeochemical variables of BFM are provided by the CMEMS reanalysis of Mediterranean Sea biogeochemistry (period 1999-2015, Teruzzi et al., 2014) produced by the MedBFM model system. The initialization profiles are extracted from the MedBFM model output array, taking the nearest model point to the BGC-Argo position in time and space. The vertical profile of eddy diffusivity coefficient $D_v(z)$ is here represented as a Gaussian-shaped function, using potential density values for calculating the mixed layer depth (MLD) with a

density-based criterion (D'Ortenzio and Prieur, 2012). The Gaussian shape is chosen for simplicity to allow a gradual increase



| SIM | MODELS DESCRIPTION |
|---|---|
| REF | Reference - $E_{d\,PAR}$ from Bio-Argo Floats ; $D_{vbackground} = 10^{-4} m^2/s$ |
| CL1 | as REF with continuous daily light |
| CL2 | as REF with continuous daily light and $D_{vbackground} = 10^{-6} m^2/s$ |
| MLD1 | as REF with $D_{vbackground} = 5\,10^{-5} m^2/s$ |
| MLD2 | as REF with $D_{vbackground} = 10^{-5} m^2/s$ |
| MLD3 | as REF with $D_{vbackground} = 5\,10^{-6} m^2/s$ |
| MLD4 | as REF with $D_{vbackground} = 10^{-6} m^2/s$ |
| OPT1 | Riley formula: $K_{d\,PAR} = 0.04 + 0.054\,Chl^{2/3} + 0.0088\,Chl$ |
| OPT2a | |
| OPT2b | $K_{dPAR} = a\,Chl^b + c$ |
| OPT2c | |
| OPT2d | |
| OPT3 | $K_{d\,PAR}$ for the first optical depth $z_{opt} = z_{eu}/4.6$ |
| OPT4a | as OPT2a + Chl degradation to CDOM - time scale 1 day |
| OPT4b | as OPT2a + Chl degradation to CDOM - time scale 1 week |
| OPT4c | as OPT2a + Chl degradation to CDOM - time scale 1 month |
| OPT5 | as OPT2a + CDOM following Dutkiewicz et al. (2015) |

**Table 1.** Model configurations considered in the present work, details on the parameters shown in the table are detailed in the text. All simulations include diurnal variability save for the continuous light cases (CL1 and CL2), which use 24-hour averaged irradiance.

| $K_{dPAR} = a\,Chl^b + c$ | $z_{max}$ | $R^2$ | a | b | c |
|---|---|---|---|---|---|
| OPT2a | 150 | 0.53 | $0.075 \pm 0.0015$ | $0.572 \pm 0.018$ | $0.027 \pm 0.001$ |
| OPT2b | 75 | 0.61 | $0.064 \pm 0.0015$ | $0.615 \pm 0.021$ | $0.040 \pm 0.002$ |
| OPT2c | 45 | 0.71 | $0.077 \pm 0.002$ | $0.469 \pm 0.021$ | $0.034 \pm 0.002$ |
| OPT2d | 30 | 0.75 | $0.088 \pm 0.003$ | $0.406 \pm 0.023$ | $0.029 \pm 0.003$ |

**Table 2.** Parameters derived for the optical models using the BGC Argo float data. For each version of OPT2 only data shallower than $Z_{max}$ were used to compute the regression.

of vertical mixing through the pycnocline. The whole approach and the impacts of using different parameterizations to reconstruct the mixing along the water column are shown and discussed in section 2.2.1. Since the surfacing of BGC-Argo floats is programmed at around local noon, the variability related to diurnal variation of solar irradiance is accounted according to Kirk (1994).





### 2.2.1 Vertical Mixing Models

Unlike in the case of radiometric data, we have only access to indirect information on vertical mixing dynamics, which has been described in terms of potential density obtained from temperature and salinity data along the water column. The vertical eddy diffusivity coefficient ($D_v$) is defined as a Gaussian-shaped function in the form of:

$$D_v = D_{vMLD}e^{-0.5(z/(\sigma MLD))^2} + D_{vbackground} \tag{1}$$

$\sigma$ equals 0.3 in all simulations, and was identified after initial tuning. The values in the REF model equal $D_{vMLD}$=1.0m$^2$/s and $D_{vbackground}$=10$^{-4}$m$^2$/s

The mixed layer depth (MLD) was defined with the density criterion at the threshold value (D'Ortenzio and Prieur (2012)):

$$\Delta\rho_\theta = |\rho_\theta(10m) - \rho_\theta(z)| = 0.03kg/m^3 \tag{2}$$

In the simulations MLD1, MLD2, MLD3, and MLD4, we perturb the $D_{v\,background}$ values for two orders of magnitude (from $10^{-6}$ to $10^{-4}$ $m^2/s$) in order to estimate the impact that such variations have on the shape of the modeled chlorophyll profile compared to the measured one (see 1).

### 2.2.2 Bio-Optical Models

In addition to the reference optical model (REF), which is based on radiometric measurements from BGC-Argo floats, several alternative solutions were considered. They differ in the method used to evaluate $K_{dPAR}$, which is parametrized as a function of Chl and/or CDOM concentration rather than being directly calculated from BCG-Argo irradiance data (Tab.1 and Tab.2). OPT1 uses the relationship obtained by a statistical analysis done by Riley (1956, 1975):

$$K_{dPAR} = 0.04 + 0.0088[Chl] + 0.054[Chl]^{2/3} \tag{3}$$

In model OPT2 we derived a statistical regression between $K_{dPAR}$ and Chl $a$ measured by BGC-Argo floats at four different depth ranges: 150 m, 75 m, 45 m and 30 m (OPT2a to OPT2d, see Tab.2 for details):

$$K_{dPAR} = a + b[Chl]^c \tag{4}$$

with a and b defined as the regression coefficients and c as the exponent (values reported in Tab.2; the confidence intervals were calculated with the Student's two-sided t-test, where the significance level $\alpha$ was set equal to 0.05). The diffuse attenuation coefficient $K_{dPAR}$ was calculated for PAR measured by the BGC-Argo floats as the local slope of [ln(Ed)] for layers of 15 m thickness for the euphotic depth range. The euphotic depth corresponds to an attenuation of downward planar irradiance to 1% of the subsurface value (Kirk, 1994).

Albeit the regression based on the upper 30 m depth range measurements showed the highest correlation, we considered all four bio-optical models and adopted them in simulations OPT2a,b,c and d.





In model OPT3, based on the BGC-Argo data set, $K_{dPAR}$ is calculated for the first optical depth (Morel, 1988), the layer of interest for satellite remote sensing (Gordon and McCluney, 1975), and then adopted as a constant parameter for the entire water column. This kind of description of the diffuse attenuation coefficient has been used also in the 3-dimensional version of the OGSTM-BFM model, which integrates data from satellite sensors $K_{d490}$ as the external optical forcing in the Beer-Lambert

formulation (for more details see Lazzari et al., 2012, section 2.2.3).

The OPT4 and OPT5 models attempt to include the effect of CDOM as it can account for 50% of the light absorption budget in the Mediterranean Sea (Organelli et al., 2014; Morel and Gentili, 2009a). In OPT4 the assumption is that the degradation of chlorophyll delays the decay of phytoplankton (Organelli et al., 2014). The attenuation of light is therefore affected by a progressive accumulation of such a constituent ("dead" chlorophyll, initialized at zero concentration) and the accumulation

is compensated by a decay (first order kinetic) that is set at different e-folding characteristic times: 1 day (OPT4a), 1 week (OPT4b) and 1 month (OPT4c). In OPT5 we used the formulation of CDOM dynamics as described in Dutkiewicz et al. (2015): a 2% fraction of all the dissolved organic matter (DOM) fluxes is directed to CDOM, including both temperature related decay and a photodegradation term based on PAR (Bissett et al., 1999). Additional investigations are provided in section 3.3 to discuss the dynamics of CDOM along the water column. Given the mono-spectral formulation presently used, the attenuation

of CDOM on PAR is computed by averaging the exponential law of CDOM absorption on the visible range.

## 3    Results and Discussion

Following the objectives of the paper, we considered four classes of simulations which correspond to the following subsections: the reference simulation, a subset with perturbed vertical mixing models, a subset of tests with different optical models, and a last group of additional tests involving CDOM description and diurnal cycle. The outputs are validated qualitatively and

quantitatively in terms of the profile shapes and the depth of the deep chlorophyll maximum (DCM). The DCM definition is based on the absolute maximum of chlorophyll along depth, excluding results of DCM shallower than 40 meters or deeper than 200 m, as well as the ones with concentrations lower than 0.1 $mg/m^3$. All results, both for model and BGC-Argo floats, are averaged on a weekly basis. The model outputs are finally compared by means of match-up diagrams and Target and Taylor diagrams (Jolliff et al., 2009). For a couple of simulations (REF and CL1), skills are also compared at the surface layer (0 - 25

m).

### 3.1    Reference Simulation

The overall model skill in the REF configuration is shown in Fig.2. A good correlation (r=0.81, p-value<0.005) is obtained between DCM depth derived from BGC-Argo floats and the modeled one. The residual plot indicates that the deviation is normally distributed (Fig.2, incut panel). The DCM depth range varies typically between 50-70 m in western areas (ALB, SWM1,

SWM2, NWM, TYR) and is generally deeper in eastern areas (ADR2, ION3, LEV2, LEV3), between 100-140 m. Model tends to slightly underestimate the DCM variability (Fig.2, regression slope = 0.79 < 1), in fact, deeper simulated DCM are around 125 m depth whilst floats data reach 140 m as measured by the lovbio18c BGC float (WMO code 6901528) deployed in the

LEV3 subbasin.

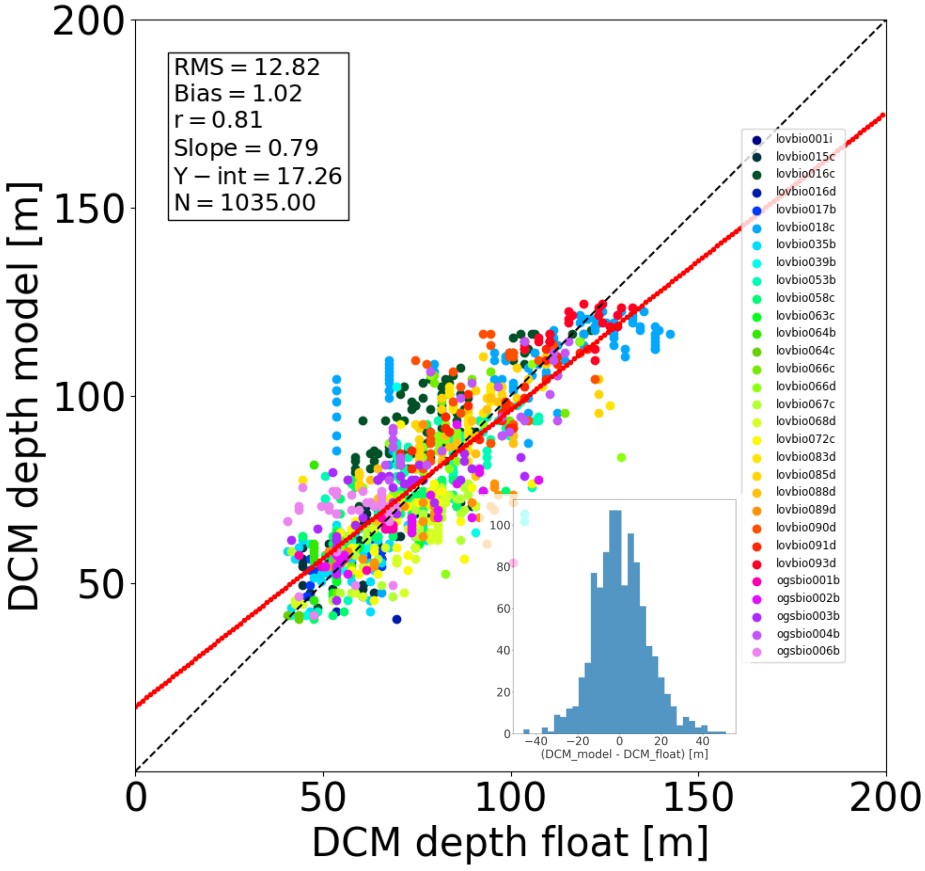

**Figure 2.** Match-up diagram comparing the depth of DCM obtained from BGC-Argo floats data versus REF model results. Each dot corresponds to a weekly profile. DCM depth definition is detailed in text. The red line is the linear regression of the data versus model defined by slope and Y-int reported in the box. RMS, Bias and Y-int reported in the top left box are in meters. The correlation r is significant, p-value<0.005. The incut figure shows the histogram of the residuals.

The chlorophyll patterns present a high variability both at temporal and vertical scales, Fig.3 to Fig.6. The subsurface chlorophyll pattern is formed by patchy structures and when fully stratified conditions occur, it is generally deeper moving eastward.

5  BGC-Argo observations indicate that DCM is further eroded by the vertical mixing occurring generally in autumn or early winter. Simulations provide an adequate reproduction of the timing of the chlorophyll mixing and therefore the DCM erosion. The model reproduces also the vertical chlorophyll distribution of the following stratification period (i.e. summer). In fact, if we compare point-to-point all the Hovmoeller maps (depth and time variability) for measured and simulated chlorophyll (examples are reported in Fig.3, Fig.4, Fig.5, Fig.6) we obtain a significant average correlation of 0.75. This confirms quantitatively

10  that the alternation of mixing and stratification phases, as seen from BGC-Argo chlorophyll measurements, is well reproduced. At surface, the increase in chlorophyll is triggered by rather shallow mixing (0-75 m layer). Overall, results indicate that the





discrepancies between the model and data are higher not only before mixing, but also in the initial phase of the water column re-stratification, hence the transient phases before and after mixing are rather critical to simulate.

In addition to the correct timing of the alternation of mixing and stratification phases, proved by the good correlation, the simulated chlorophyll also reproduces episodic signals, such as the deepening of chlorophyll due to specific mixing events. For example, the mixing event in the NWM sub basin, reaching approximately 200 m depth during winter 2015, triggers an intrusion of chlorophyll (approximately $0.2 mgChl.m^3$) in the deeper layers consistently to BGC-Argo float measurements (BGC float lovbio067c, Fig.3). Similar dynamics is reproduced in winter 2014 (Fig.4), for the lovbio035b BGC float drifting from NWM toward ALB sub-basin.

Considering the float trajectories, two kinds of situations are possible: the BGC-Argo float trajectory is relatively stationary in the deployment area (as shown in Fig.3, 5 and 6), or the float passively migrates extensively, following a given water mass (as in Fig.4). It appears that also in the second case, when lateral dynamics effects could be important in BGC-Argo float measurements, our approach allows to represent the measured chlorophyll patterns adequately. However, it should be noted that in the present multi-float simulation there are no trajectories that include both west and east Mediterranean basins. In this case, strong gradients between deep water nutrient inventories could invalidate the approach, and hence a nudging or a more sophisticated technique could be required (Kohlmeier and Ebenhöh, 2009). The lateral advection processes could indeed play an important role, although it appears that in the present case considering data-driven mixing and turbulence effects allows to simulate correctly the seasonal variability. The REF simulation can be therefore used as a reference for the tests on mixing and bio-optical models analyzed in the following sections The results of the REF simulation show that irradiance along the water column, besides mixing, is the driving mechanism that controls the DCM depth. As shown in Fig.7a, there is a significant correlation between DCM depth and the depth of $1\%$ surface PAR (euphotic depth), both in the case of the measured Chl and in the simulated one. Similar results, valid on annual average conditions, were found by Mignot et al. (2014) in their Eq. 9, where euphotic depth results to be $0.73\%$ rather than $1\%$ of surface PAR. But the prevailing interpretation in Mignot et al. (2014) was that the DCM is located at fixed PAR value oscillating approximately near the $0.5 \ \mathrm{molquanta.m^{-2}.day^{-1}}$ isolume. Similar conclusions can be derived in the present work, Fig.7b, where we show that same results are valid both in the case of the BGC Argo float and model with a higher variability of critical PAR values in the case of shallower DCM. There are exceptions: in Fig.8, which shows the evolution of a BGC-Argo float during a five-week period, the model-predicted DCM seems strongly constrained by light regime whilst observed DCM fluctuates up and down over the euphotic depth (see transition from "T=29 weeks" to "T=31 weeks").

The Mediterranean Sea is considered as a nutrient-limited basin (e.g. Crispi et al., 2001; Lazzari et al., 2016; Powley et al., 2017), therefore an insight on the role played by nutrients requires further investigation. Phosphate dynamics evidences how the increase in surface chlorophyll is driven by mixing in the surface layers of nutrients. During the stratification period, the phosphocline follows the euphotic layer threshold. From results shown hereby, it appears that beside strong correlation between light and the DCM depth, nutrient concentration is an important driver that regulates the phytoplankton biomass at DCM, as depicted in Fig.9, where the western basin exhibits significantly higher values, both of phosphate and biomass, compared to the eastern one.



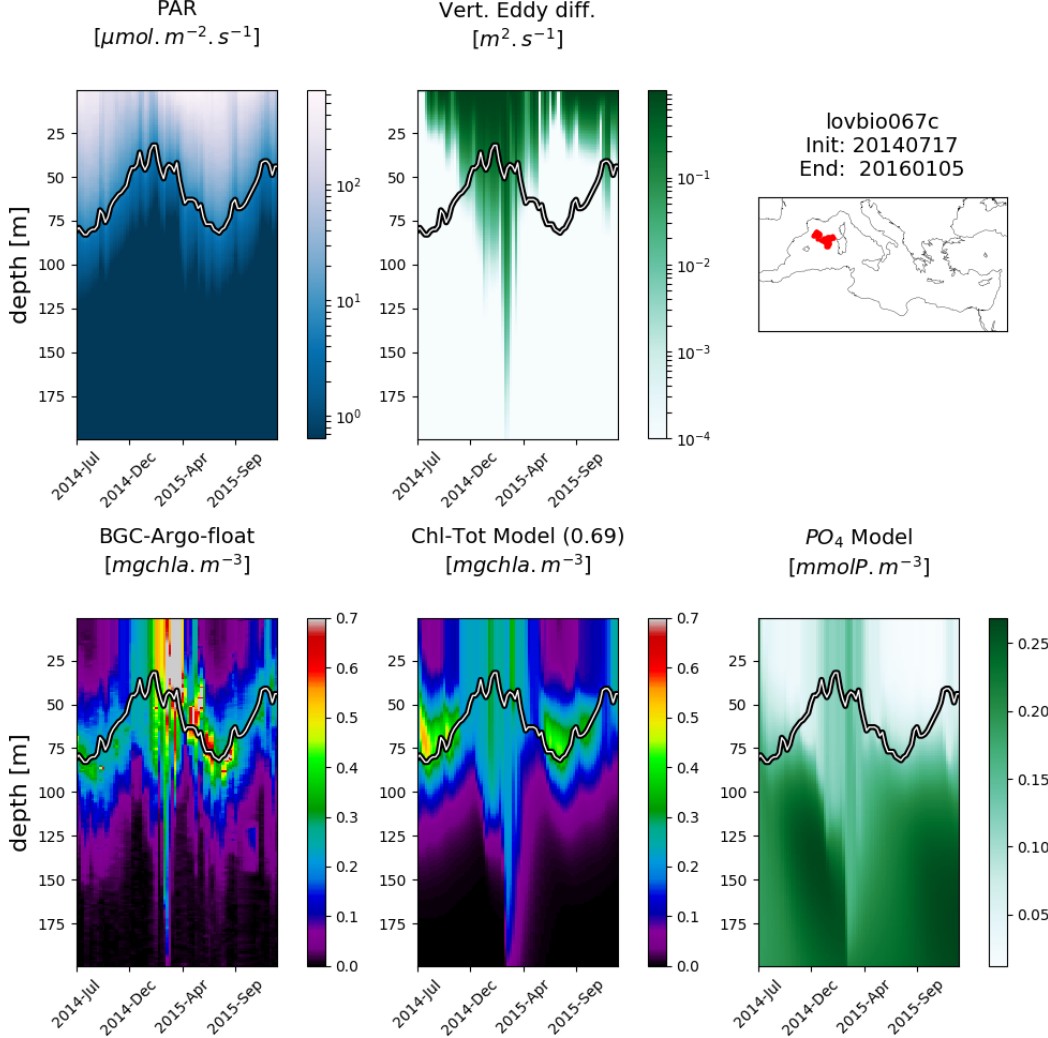

**Figure 3.** Hovmoeller diagrams of BGC float lovbio067c (WMO code 6901649) comparing measured results and simulated ones (REF). The 6-imaged composite is organized as follows: top row shows PAR, vertical eddy diffusivity and the float trajectory; bottom row shows chlorophyll derived from fluorescence measurements, simulated chlorophyll and simulated phosphate. The thick black-white line indicates the depth where PAR has values of $0.5 \, \mathrm{molquanta.m^{-2}.day^{-1}}$ (Mignot et al., 2014). The number in parentheses in Model Chl indicates point-point correlation with BGC-Argo float chlorophyll.



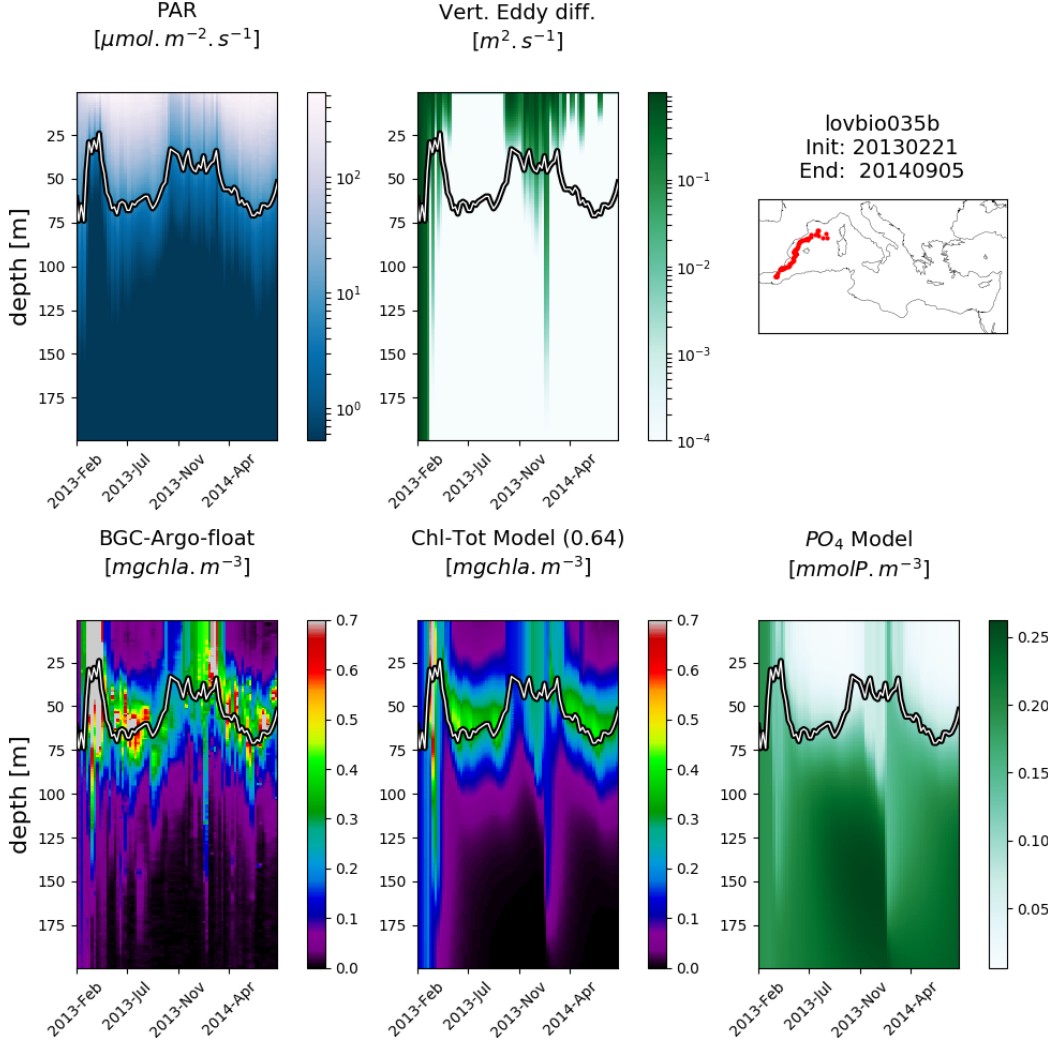

**Figure 4.** Hovmoeller diagrams of BGC float lovbio035b (WMO code 6901511) comparing measured results and simulated ones (REF). The 6-imaged composite is organized as follows: top row shows PAR, vertical eddy diffusivity and the float trajectory; bottom row shows chlorophyll derived from fluorescence measurements, simulated chlorophyll and simulated phosphate. The thick black-white line indicates the depth where PAR has values of $0.5 \ \mathrm{molquanta.m^{-2}.day^{-1}}$ (Mignot et al., 2014). The number in parentheses in Model Chl indicates point-point correlation with BGC-Argo float chlorophyll.





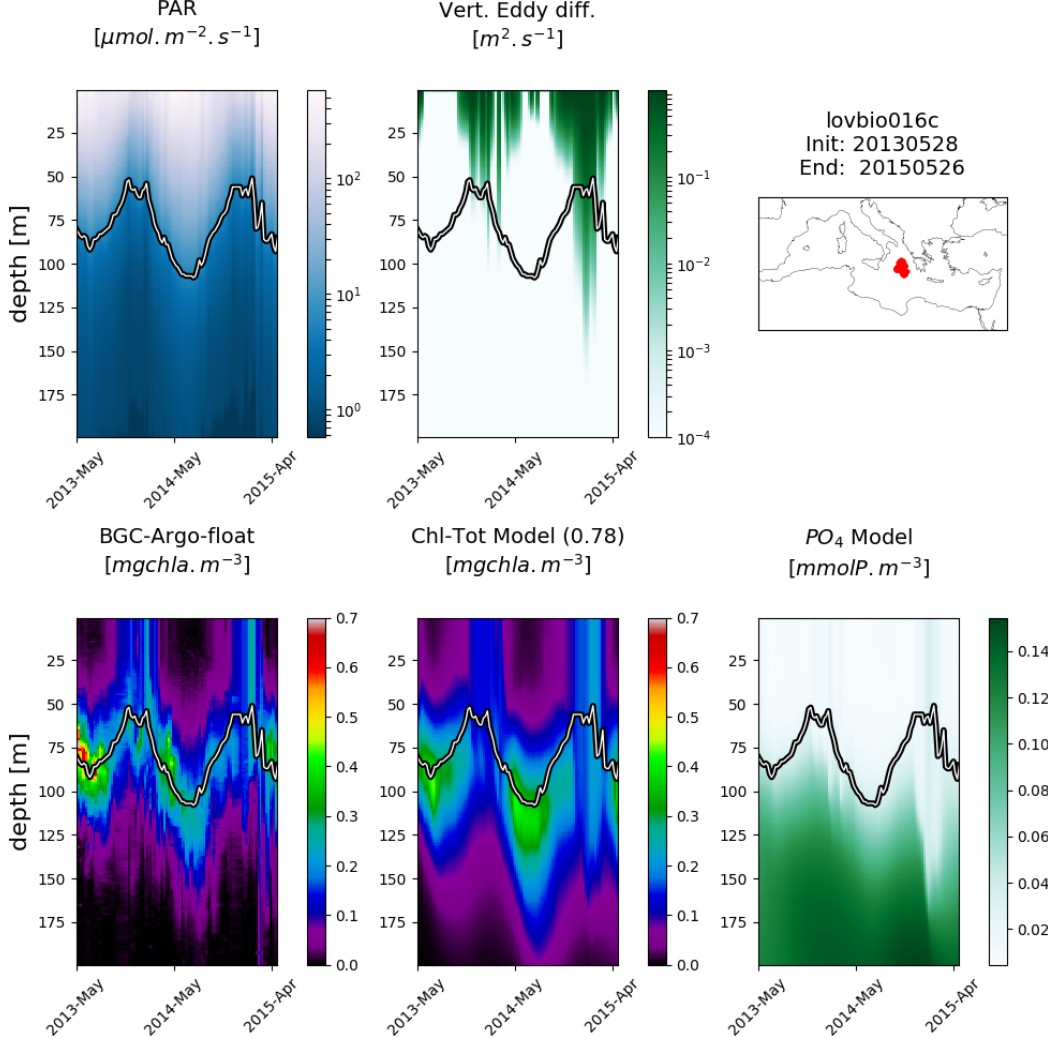

**Figure 5.** Hovmoeller diagrams of BGC float lovbio016c (WMO code 6901510) comparing measured results and simulated ones (REF). The 6-imaged composite is organized as follows: top row shows PAR, vertical eddy diffusivity and the float trajectory; bottom row shows chlorophyll derived from fluorescence measurements, simulated chlorophyll and simulated phosphate. The thick black-white line indicates the depth where PAR has values of $0.5 \; \mathrm{molquanta.m^{-2}.day^{-1}}$ (Mignot et al., 2014). The number in parentheses in Model Chl indicates point-point correlation with BGC-Argo float chlorophyll.





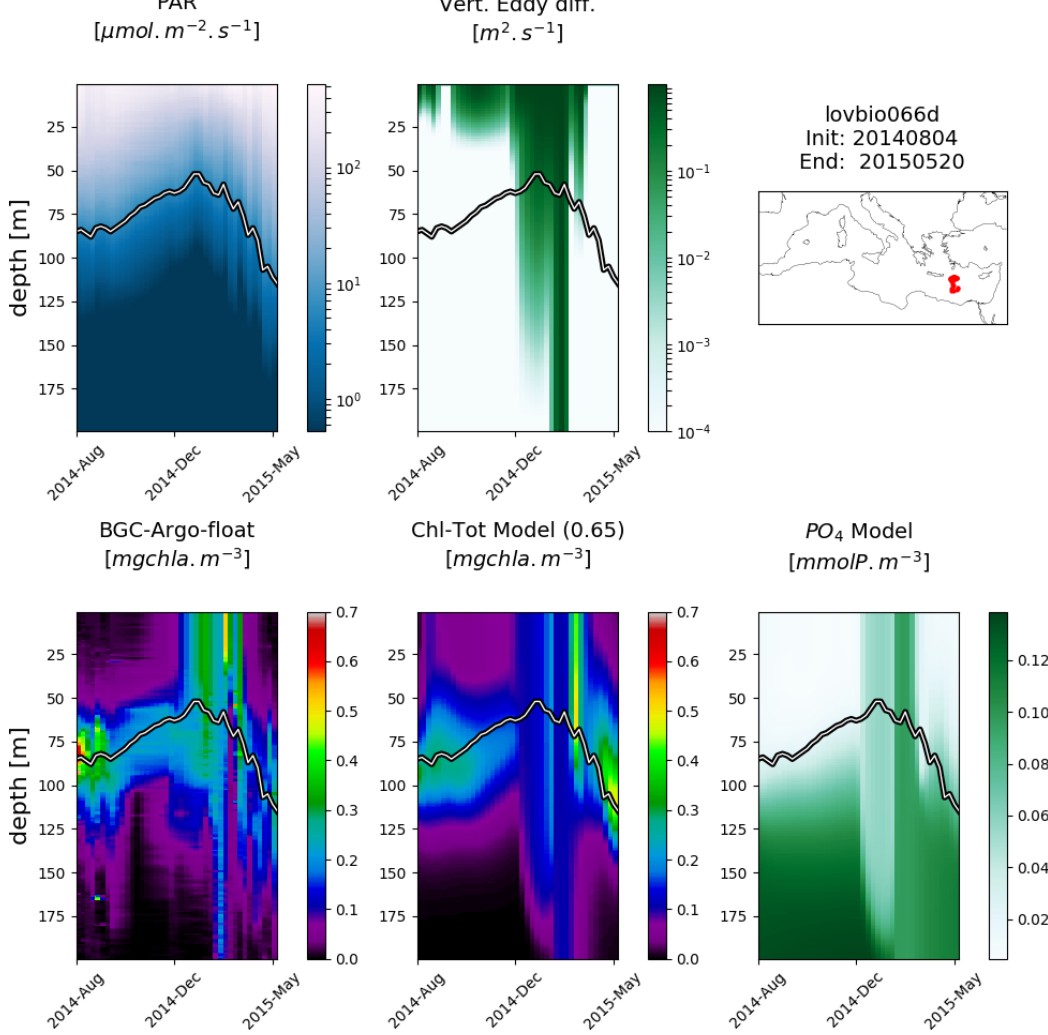

**Figure 6.** Hovmoeller diagrams of BGC float lovvio066d (WMO code 6901655) comparing measured results and simulated ones (REF). The 6-imaged composite is organized as follows: top row shows PAR, vertical eddy diffusivity and the float trajectory; bottom row shows chlorophyll derived from fluorescence measurements, simulated chlorophyll and simulated phosphate. The thick black-white line in the PAR panels indicates the depth where PAR has values of 0.5 $\mathrm{molquanta.m^{-2}.day^{-1}}$ (Mignot et al., 2014). The number in parentheses in Model Chl indicates point-point correlation with BGC-Argo float chlorophyll.





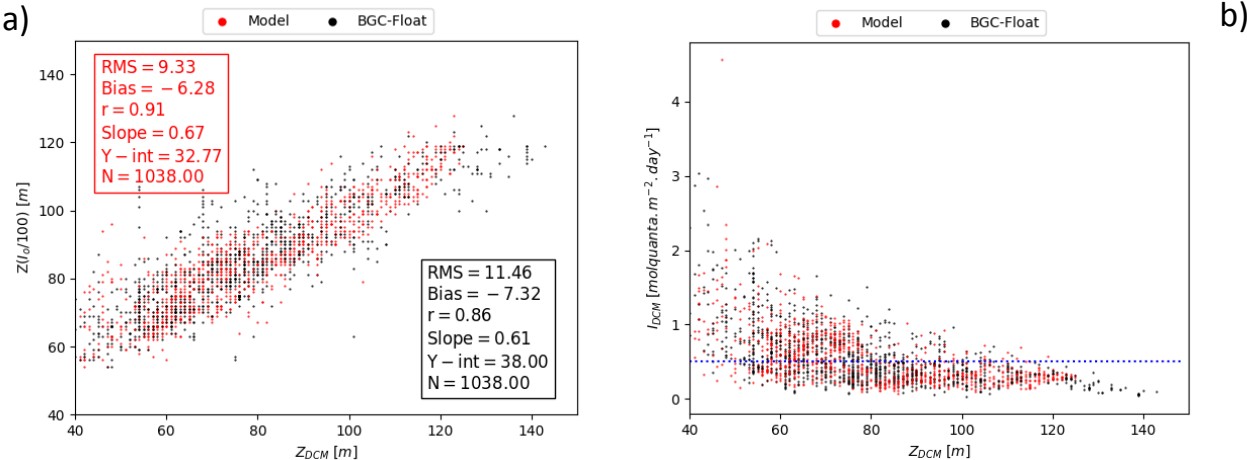

**Figure 7.** Panel a) DCM depth ($Z_{DCM}$, x-axis) compared to the euphotic depth $Z_{EU}$ ($Z(I_0/100)$, y-axis) both for model results (red dot) and measured results (black dot). Red box (top left) reports statics for model ZDCM versus Zeu, whereas the black box (bottom right) shows statistics for $Z_{DCM}$ derived from chlorophyll data versus $Z_{EU}$. Panel b) Irradiance values (y-axis) at DCM depth (x-axis) both for model results (red dot) and measured results (black dot). Horizontal blue line marks the 0.5 irradiance threshold (units $\mathrm{molquanta.m^{-2}.day^{-1}}$) as identified in Mignot et al. (2014).





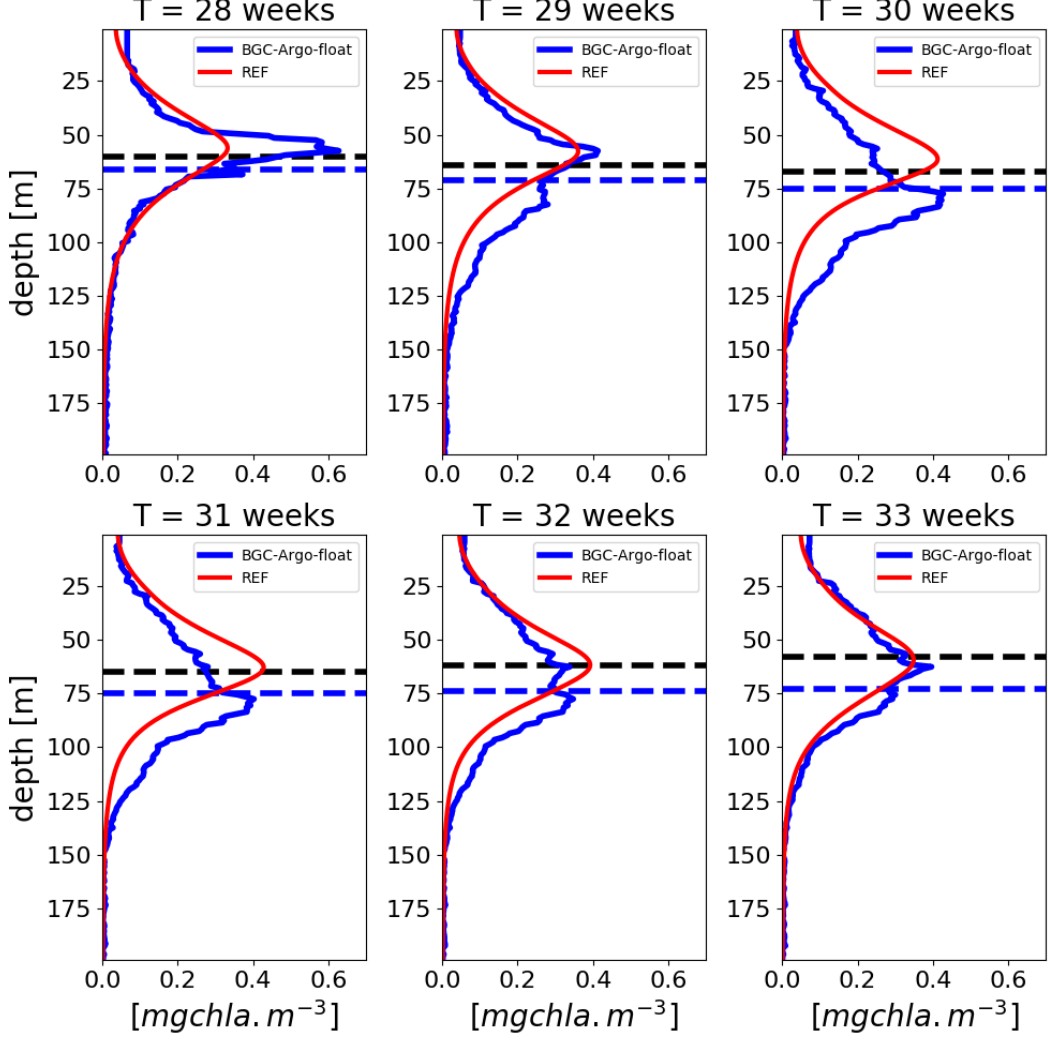

**Figure 8.** Example of a weekly time series of vertical profiles referred to the REF simulation of lovbio035b BGC-Argo float (Fig.4) and compared to BGC-Argo float chlorophyll (thicker line). The horizontal dashed blue line represents the euphotic depth, defined as $I_0/100$, where $I_0$ is the surface irradiance. The horizontal dashed black line indicates the depth where measured PAR has a value of 0.5 molquanta.m$^{-2}$.day$^{-1}$ as identified in Mignot et al. (2014).



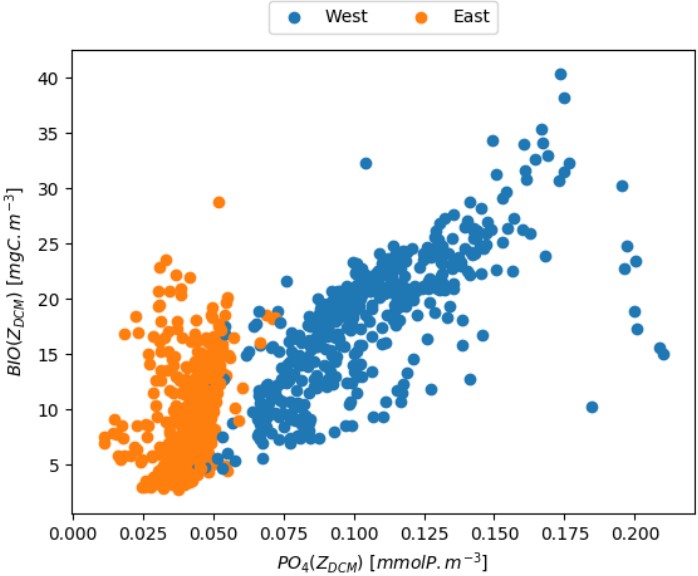

**Figure 9.** Phosphate concentration (x-axis) and total biomass concentration (y-axis) of phytoplankton at DCM depth. All the modeled BGC float trajectories are included.

## 3.2 Vertical Mixing Models

As shown in the previous section, the vertical distribution of chlorophyll displays a distinct variability, which can be at least partially ascribed to mixing (i.e. vertical eddy diffusivity). Typically, higher vertical eddy diffusivity values imply smoother structures. During the stratification phase, when DCM forms, the controlling mixing parameter is the background diffusivity

5   $D_{\text{v background}}$. Simplified theoretical models, such as the KiSS (after the names of: Kierstead and Slobodkin, 1953; Skellam, 1951), can provide rough quantitative scales in order to determine the minimum vertical length scale ($L_0$) that allows formation of stable biomass patches, including the DCM, in a steady state hypothesis:

$$L_0 \propto \sqrt{\frac{D_v}{\mu}} \qquad (5)$$

where $D_v$ is the vertical diffusivity coefficient and $\mu$ is the growth rate; in stratified conditions, $D_v = D_{\text{vbackground}}$. Consider-

10   ing any compact vertical interval with favorable conditions for plankton growth (in terms of irradiance and nutrient availability), the increase of background diffusion over a critical value will produce a dispersal of patchy structures (e.g. a relative maximum of chlorophyll concentration), whereas an increase in growth rate $\mu$ can drive the formation of finer scale structures by a reduction of $L_0$.

The dynamics presented in this study are much more complex compared to KiSS, both in BGC floats data and in the 1-D

15   medium-complexity biogeochemical model (BFM). Vertical eddy diffusivity can simultaneously affect nutrients, phytoplank-



ton, and mesozooplankton with intricate interactions and feedbacks, which in turn make difficult to derive analytical solutions. Moreover, unlike KiSS, the model and the environment are hardly ever in a steady state condition, as a result of daily and seasonal oscillations in the physical forcings, which are essentially due to variability in diel irradiance and vertical mixing.

5    Several simulations, labelled as MLD1, MLD2, MLD3 and MLD4, were carried out by changing the background vertical eddy diffusivity coefficient $D_{\mathrm{vbackground}}$ values by two orders of magnitude (from $10^{-6}\mathrm{m}^2/\mathrm{s}$ to $10^{-4}\mathrm{m}^2/\mathrm{s}$, see Tab.1). This subset of simulations (with float-derived PAR) clusters at a correlation of approximately 0.8 with a root mean square difference (RMSD) of DCM depth between 10-15 m. Modeled chlorophyll profiles appear much smoother than the observed ones, following a Gaussian shape for all tested values of eddy diffusivity. Small scale patterns are not detectable even when $D_{\mathrm{vbackground}}$
10   values are reduced to a minimum. Further analyses concerning these aspects are shown in section 3.4.



## 3.3 Bio-Optical Models

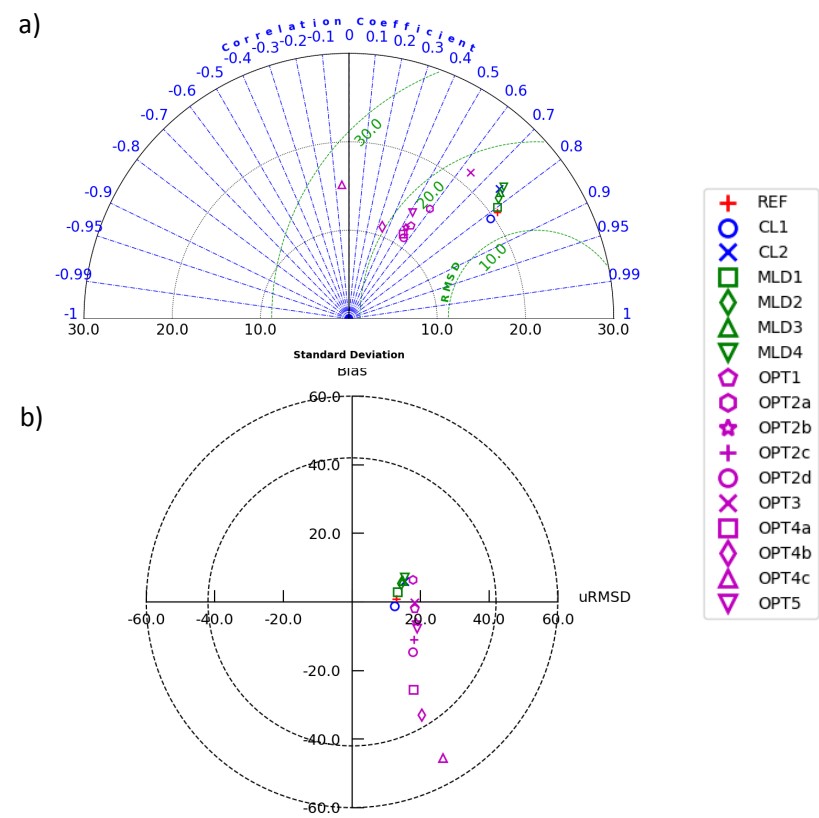

**Figure 10.** Panel a): Taylor diagram showing the model skill in reproducing DCM depth as compared to data. Correlation is represented by the angle with positive x axis, whereas the distance from the origin depicts the standard deviation. Green circles illustrate iso-contours of RMSD levels. Panel b): Target diagram showing the model skill in reproducing DCM depth compared to data. Distance to the origin is the RMSD, all units are in meters. The position on the x axis is positive if the standard deviation in the model is higher than data and negative in the opposite situation. For the sake of completeness, all the model considered are reported in these summary skill diagrams.

The adoption of alternative bio-optical models (OPT1, OPT2, OPT3) results in a correlation reduction from 0.8 (of the REF simulation) to 0.6-0.5 (Fig.10). In particular, OPT3, with almost zero bias, displays an intermediate skill compared to assimilated PAR simulations (REF and the OPT1 and OP2 bio-optical models). In general, the OPT1 and OPT2 cluster of models
5 show slightly lower correlations with a RMSD of approximately 20 m in all cases, with an increasing bias (almost zero for OPT1 and from 6 m (OPT2a) to -14 m (OPT2d)). This may stem from the fact that the DCM depth statistics performed for




the OPT2a to OPT2d models ranged from 150 m to 30 m respectively, therefore lowering the number of points considered due to a reduced depth interval. Despite an increasing correlation of the bio-optical model linear regression with decreasing depth range, it should be underlined that the equations for lower depth ranges (such as OPT2d for the first 30 m) most likely do not perform well at greater depths, hence a higher bias despite of a higher correlation coefficient (R, p-value < 0.005). The

ensemble of simulations with alternative optical models shows in all cases smoother curves compared to the measured chlorophyll profiles (see Fig.11). The self-shading effect increases from OPT2a to OPT2d, as explained above, due to the different depth range of the dataset used to compute the bio-optical algorithm regression. Some of the bio-optical models considered in the present work, in particular OPT1, OPT2a and OPT2b, appear to be able to reproduce the DCM depth gradient between western and eastern sub-basin with a tolerance of +/- 10 meters (Fig.12). In the previous publications (Crispi et al., 2002;

Lazzari et al., 2012), the correct simulation of DCM depth longitudinal gradient was obtained by forcing the system with a space-time dependent light attenuation parameter based on a Secchi disk climatology or on satellite Kd490 data. Both the empirical approaches prevent to understand whether the origin of such gradients is directly related to the external forcings or, on the contrary, if it can be interpreted as a self-emerging property. Here the interpretation of self-emerging property is related to the emergence of a feature not directly and explicitly imposed from the choice in the boundary conditions or from the choice

of the model parameters used for the numerical experiment (de Mora et al., 2016). Results shown in Fig.12 suggest that a gradient in DCM depth could be partially reproduced and explained in terms of internal biogeochemical processes and partially due to external forcings (e.g downward irradiance and nutrient initial conditions), even without considering lateral dynamics. In fact, the average surface PAR of the dataset we considered is higher in the eastern areas, especially during the months of January (40%), September (15%), October (22%), November (36%), December (16%), probably due to atmospheric weather

winter conditions. During summer, when the DCM stabilizes, the west-east differences in measured surface PAR are lower and they oscillate around 10%, but still they contribute in increasing irradiance penetration in deeper layers. The western and eastern basins are also different in terms of nutrient regimes that in turn impact on biogeochemical dynamics and on the DCM depth gradient in non-trivial ways. Another key factor pertains the shorter wavelengths (400-450 nm) in the visible spectrum: when light penetrates deeper along the water column, compounds like CDOM are more effective in absorbing light in the blue

spectrum and might in turn enhance spatial gradients in irradiance regimes. These factors could synergistically contribute to a deeper DCM in the eastern sub-basins. The monospectral formulation of the present model cannot address this aspect, but future model versions equipped with multi-spectral bio-optical models linked with specific PFT and CDOM absorption terms (Dutkiewicz et al., 2015) are a fundamental step for further investigations.

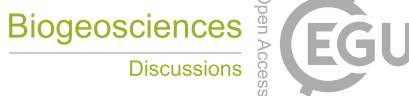

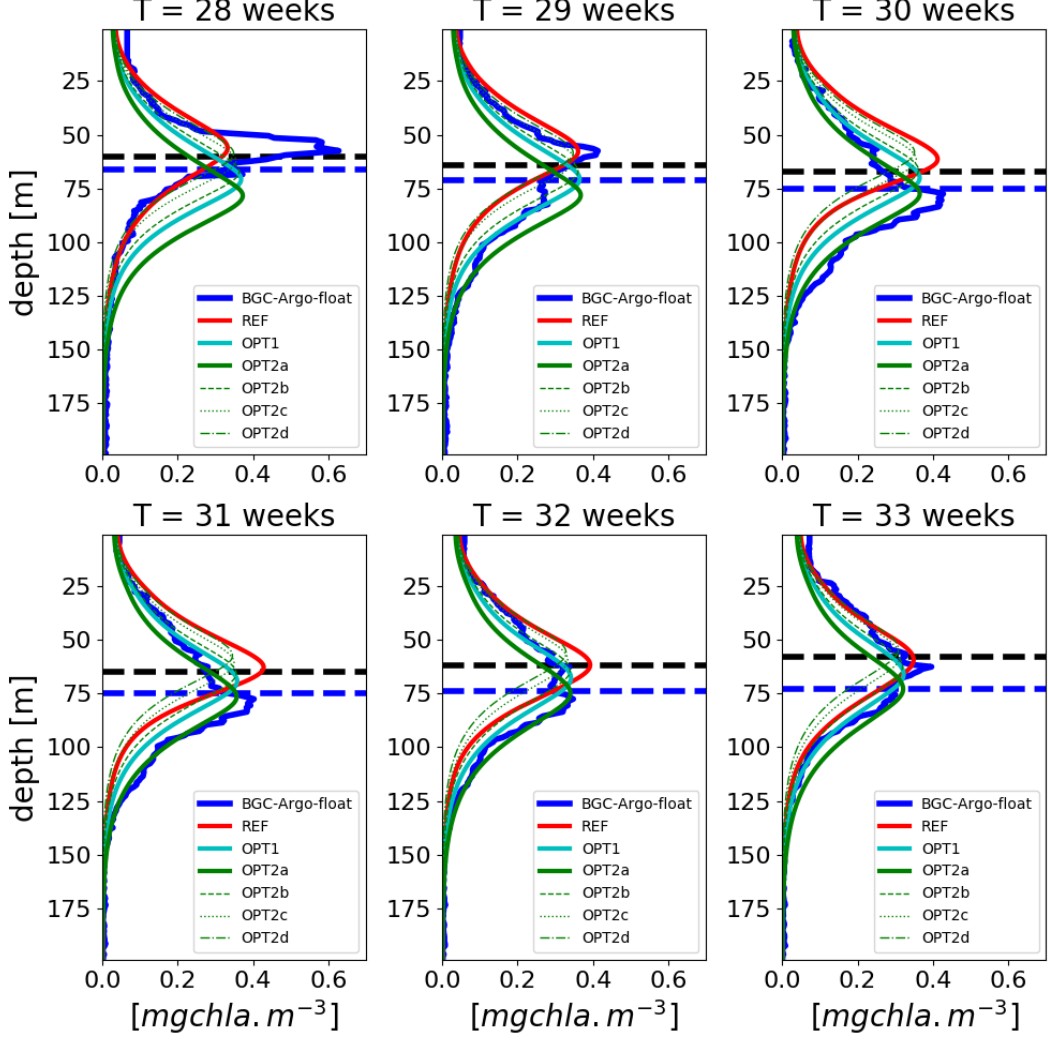

**Figure 11.** Example of a weekly time series of vertical profiles referred to lovbio035b BGC-Argo float (Fig.4) showing REF simulation and alternative bio-optical models OPT1 and OPT2, and compared to BGC-Argo float chlorophyll (thicker line). The horizontal dashed blue line represents the euphotic depth, defined as $I_0/100$, where $I_0$ is the surface irradiance. The horizontal dashed black line indicates the depth where PAR has a value of $0.5 \ \mathrm{molquanta.m^{-2}.day^{-1}}$. The legend reports the model configurations listed in Tab.1.



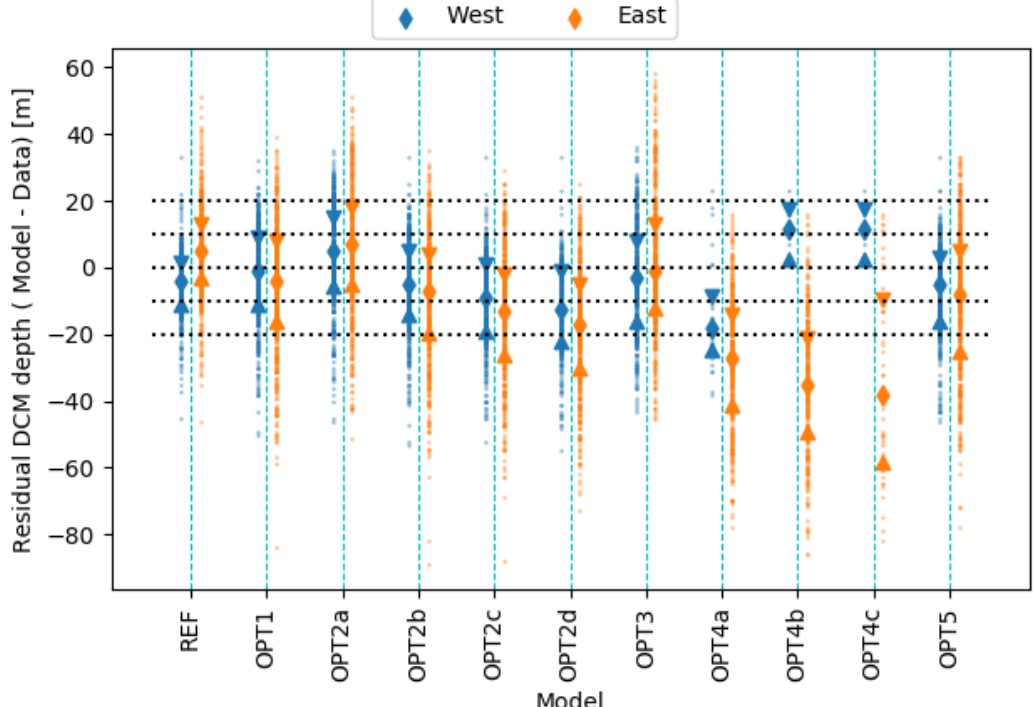

**Figure 12.** Scatter plots of the residual difference between measured and modeled DCMs. The x axis reports the model configurations listed in Tab.1. On the y-axis the median of the residuals for the west (blue) and east (orange) profiles is shown. Triangles indicate the 25th and 75th percentile.

### 3.4 Daily versus constant PAR forcings

The use of daily averaged irradiance (i.e. with continuous light, CL1 and CL2) was compared against simulations that included diurnal variability. A consistent reduction of surface chlorophyll concentrations was observed in the former case (Fig.13, with a correlation lower than REF), affecting much less (in relative terms) the values around DCM (CL2 is shown in Fig.14). Near

5 surface, phytoplankton is clearly stressed by low nutrients (and especially in the eastern part) whereas deeper, near the DCM, the trophic limitation is weaker, sometimes null (Behrenfeld and Boss, 2003; Behrenfeld et al., 2004). One possible explanation could be that the light limitation at the DCM for a low-irradiance regime is almost linear, thus the averaging effects appear to be having a smaller impact than at the surface levels, where light limitation is highly non-linear due to saturation.

Furthermore, the Geider formulation for chlorophyll acclimation (Geider et al., 1998) in the case of diurnal variability generates

10 an increase of the chlorophyll-to-carbon ratio. This aspect can have important implications in operational applications where data assimilation is employed to improve model skill. At surface, the adoption of a diurnal cycle formulation could reduce the



correction made by the assimilation scheme, and therefore minimize possible spurious trends introduced by the assimilation (Gehlen et al., 2015).

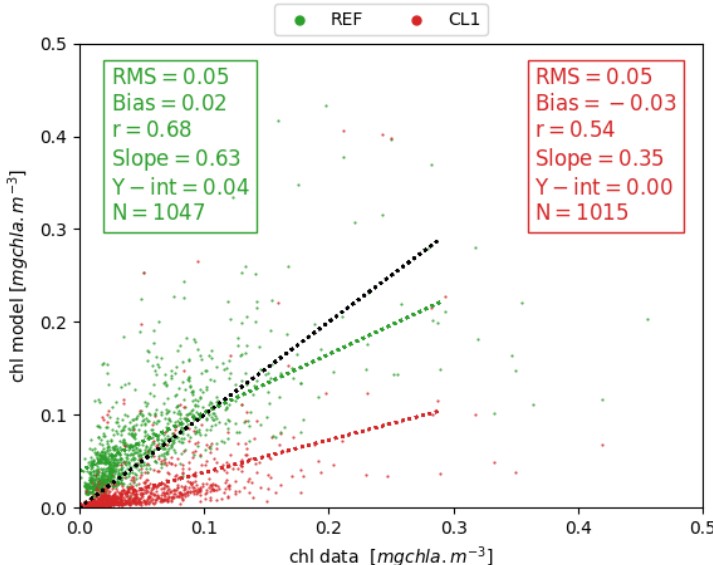

**Figure 13.** Scatter plot comparing 0-25 m average surface chlorophyll versus BGC Argo float data for the stratified period condition, DCM > 40m.



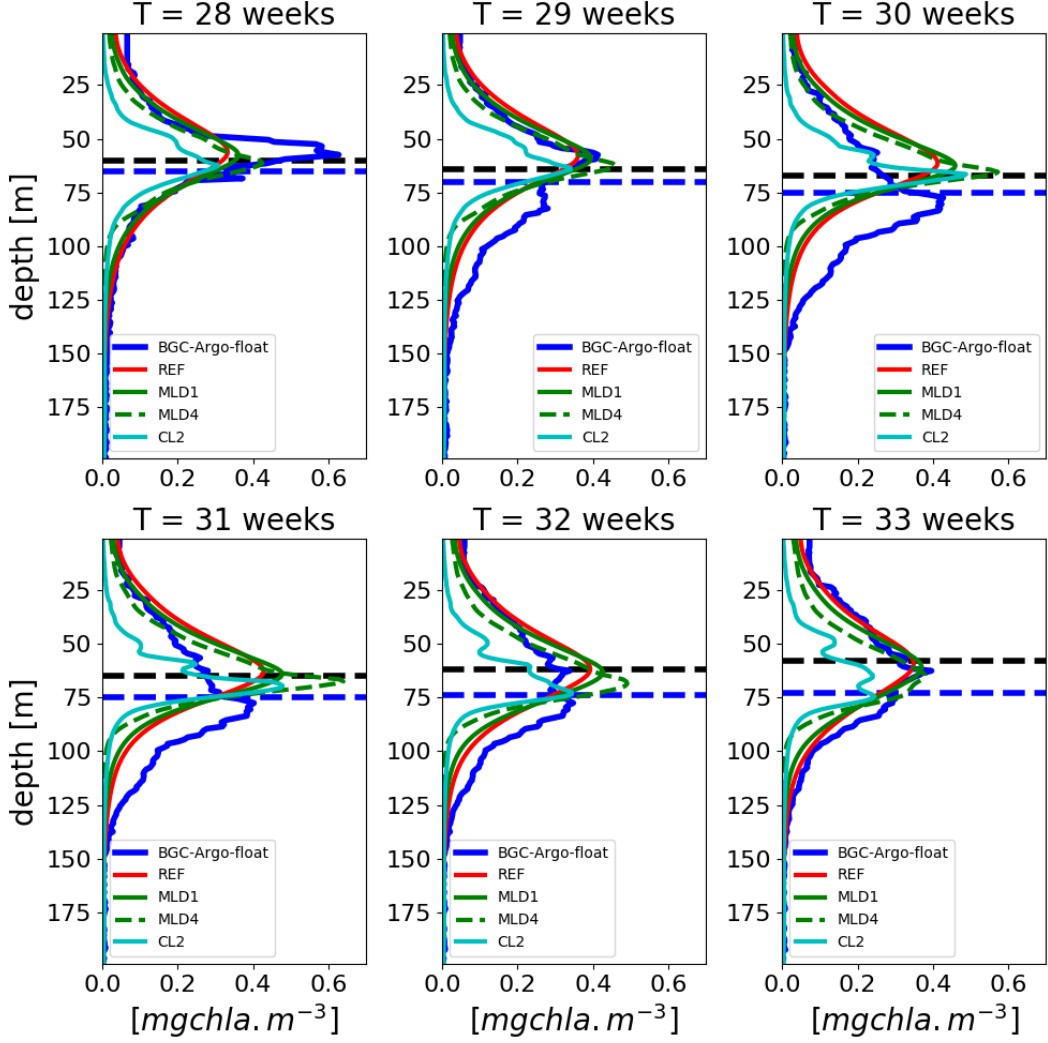

**Figure 14.** Example of a weekly time series of vertical profiles referred to lovbio035b BGC-Argo float (Fig.4) based on diel variability and constant daily light descriptions, and compared to BGC-Argo float chlorophyll (thicker blue line). The horizontal dashed blue line represents the euphotic depth, defined as $I_0/100$, where $I_0$ is the surface irradiance. The horizontal dashed black line indicates the depth where PAR has a value of $0.5 \, \mathrm{molquanta.m^{-2}.day^{-1}}$. The legend reports the model configurations listed in Tab.1.

Combining daily-averaged irradiances with lowest diffusivity rates ($D_{\mathrm{vbackground}} = 10^{-6} \mathrm{m^2/s}$, simulation CL2) results in additional relative chlorophyll maxima at surface layers (see Fig.14, panel "T = 33 weeks"), as well as in increased patchiness of the entire vertical profile. Theoretical consideration predicts different maxima along the water column on the base of the Tilman resource competition theory applied to an heterogeneous system (Ryabov and Blasius, 2011). But presently it is difficult to assess whether the patchy structures observed in data and model are, for different reasons, realistic or artefactual. Certainly, the background diffusion needed in the model simulation to maintain such structures is very low.



Within the framework of currently used mathematical formulations in the 1-D BFM model, the inclusion of diurnal variability tends to reduce the formation of fine-scaled structures. Therefore, the effects of the diel cycle could be interpreted in terms of a reduction of diel growth ($\mu$) or possibly seen as a perturbation that has an equivalent effect of an increased diffusion.

### 3.5 Bio optical models with CDOM formulation

The OPT4 and OPT5 simulations take into consideration the CDOM dynamics by adding an additional term to the OPT2a model, where the light attenuation for PAR was described only in terms of chlorophyll concentration. In OPT4a, b, and c, CDOM is parameterized as "dead" chlorophyll, changing only the rate of chlorophyll decay from 1 day to 1 month. This simplified dynamics description derives by the high correlation observed between chlorophyll and CDOM in Morel and Maritorena (2001), although it should be noted that no analysis was carried out within the dataset examined hereby due to the lack

of CDOM concentration data. In all three model configurations, the "dead" chlorophyll accumulation results in higher turbidity levels, quantified by significantly negative DCM biases (over 40 meters in OPT04c), which result in shallower DCM compared to BGC-Argo derived profiles. Furthermore, the OPT4 set correlation with floats data is generally lower than 0.6 (Fig.10). As it was indicated in the statistical analysis, the OPT4 group of experiments presents shallower DCM depth since the attenuation of chlorophyll is overestimated even when considering the fastest degradation rates. The experiment OPT5 mimics the CDOM

dynamics described in Dutkiewicz et al. (2015). A lower bias is observed compared to the oversimplified OPT04 tests (where the correlation coefficients are spanning from 0.6 to less than 0.1 for OPT04a to OPT04c respectively). OPT5 still results in a negative bias of around 10 m compared to the values from -25 m to -40 m for OPT04a to OPT04c.

In open ocean systems, at least three different mechanisms concerning CDOM entrainment in the euphotic layer are considered: lateral flux of CDOM from terrestrial waters (allochthonous origin), production of CDOM in the euphotic layer (autochthonous

origin) and bottom-up flux of CDOM from the subsurface layer not affected by bleaching (Nelson and Siegel, 2013). Fig.15 shows an example for a BGC-Argo float deployed in the North West Mediterranean sub-basin (NWM). The model, despite the initial conditions, correctly drives CDOM absorption in deeper layers to low values whilst an enhanced surface production reinforces mineralization and bleaching and thus realizes a continuum of CDOM reactivity and lability. Results of CDOM variability from the BOUSSOLE site show that CDOM absorption ranges to a maximum value of 0.07 m$^{-1}$ and indicate that

there is a delay between phytoplankton bloom and maximum in CDOM absorption (Organelli et al., 2014, fig. 3), whereas deeper layers (below 100 m) have low CDOM absorption. The dataset shown in Organelli et al. (2014) evidences that cycles of CDOM accumulation followed depletion in the upper 10 m due to photo-degradation. In our results the bleaching has a deeper effect over all the CDOM 'productive' layer (see red and blue lines, Fig.15) and subsurface maxima of CDOM are not reproduced. Additional investigations of the OPT5 model configuration can address the dynamics of the autochthonous source

and the bottom-up flux of CDOM in this region. The lack of CDOM accumulation in deeper layers for the OPT5 configuration hinders a proper analysis of the mechanism suggested in section 3.1 related to the emergence of CDOM from subsurface dark layers. Improving model dynamics calibrations could be possibly achieved by utilizing information on CDOM light absorption from BCG-Argo floats measurements (Xing et al., 2012): this analysis could be potentially useful to understand how much of the CDOM signal is autochthonous and how much allochthonous.




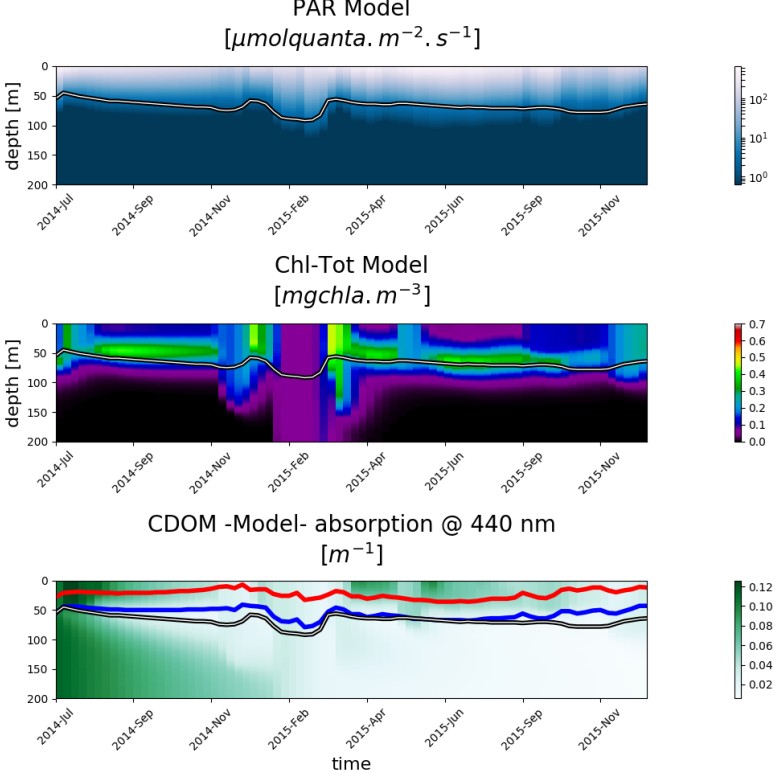

**Figure 15.** Hovmoeller diagrams for BGC Argo float lovbio068d (WMO code 6901648) showing: PAR (top), total chlorophyll (middle) and CDOM (bottom) simulated by model configuration OPT5. The white, red and blue lines depict the euphotic depth, the 100% bleaching region and the 10% bleaching depth respectively.

## 4    Conclusions

The coupled 1-D modeling/BGC-Argo observations approach presented here provides a robust and accurate reproduction of the DCM variability across the Mediterranean Sea. The model can integrate in a single framework the multi-data measurements provided by the BGC-Argo floats. DCM is a ubiquitous feature of the chlorophyll vertical structure in the Mediterranean, and

5    different forcing conditions generate geographical gradients in the DCM characteristics (i.e. shallower DCM in the western regions, deepening eastwards). Second-order features, such as impulsive vertical spikes or specific patterns observed in the BGC-Argo profiles, are also qualitatively reproduced. Our results can be summarized as:

- mixing and irradiance propagation control the chlorophyll dynamics;

- DCM position is mostly controlled by PAR, and the present work corroborates what found in Mignot et al. (2014);

10    - nutrients control the amount of biomass at DCM.



We demonstrate that vertical processes considered in the 1-D model, such as irradiance regimes and vertical mixing, allow to properly reconstruct a large part of chlorophyll dynamics, which was quantified also by the skill diagrams.

Such kind of data-rich experiments, combined with a 1-D numerical model, could be considered as a useful tool also to a broader community, rather than only to biogeochemical modelers, in particular to address process studies.

Moreover, the presented approach might be useful to quantify the amount of measured signal related to vertical dynamics and the one derived from other processes, such as horizontal advection and subduction of water masses. The usage of PAR measured from BGC-Argo floats (used in REF, CL1, CL2, MLD1, MLD2, MLD3 and MLD4) provides higher correlations compared to the configurations with alternative bio-optical models (used in OPT1, OPT2, OPT3, OPT4 and OPT5). CL1 (without diurnal cycle) shows overall the highest correlation, comparable with REF (Fig.10a).

The comparison of different bio-optical models indicates that, when lacking direct measurements of PAR in the subsurface layers, the most fitting alternatives would be the OPT3, OPT2a and OPT1, that provide a relatively lower bias and higher correlation coefficients (between 0.5 and 0.7), as well as a lower RMSD values compared to REF.

Such an analysis can also suggest the rate of improvement when considering a value of light fully integrated in the visible range of the spectrum (400 to 700 nm, REF) versus simplified approaches (e.g. all the OPT simulations here considered).

These results further support the strategic relevance of BGC-Argo data. Temperature, salinity and radiometric parameters encapsulate fundamental information for the reconstruction of primary producers dynamics and are paramount to investigate hypotheses concerning DCM formation. CDOM fluorescence data measured by BGC-Argo floats could be integrated in the simulations to further infer and reconstruct the observed biogeochemical processes.

Furthermore, considering a general 3-D biogeochemical model, it is not possible to have a full data coverage of the in-water

PAR field, therefore the present approach evaluation of bio-optical models skill is useful and the emerging considerations could be exported to more complex 3-D biogeochemical models and generalized to regions other than the Mediterranean Sea (possibly on a global scale).

*Code and data availability.*  The BFM biogeochemical model and its documentation can be downloaded at the following address: http://bfm-community.eu/. The quality-controlled databases used in the present manuscript are publicly available from the SEANOE (SEA scieNtific

Open data Edition) publisher at https://doi.org/10.17882/49388 and https://doi.org/10.17882/47142 for vertical profiles and products within the first optical depth, respectively.

*Competing interests.*  Authors declare that no competing interests are present.

*Acknowledgements.*  This work is part of the PhD project of Elena Terzić, which was funded under the CMEMS contract for the Biogeochemistry Production Unit for the Mediterranean Sea. The simulations were performed in the framework of the ISCRA C project NOVBIOGE



(HP10C8C9O6), granted by CINECA, Italy. This work was supported by the French "Equipement d'avenir" NAOS project (Novel Argo Ocean observing System), ANR J11R107-F. We acknowledge sponsorship from the MISTRALS-MERMEX project.





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
