# Peer review of "Merging bio-optical data from Biogeochemical-Argo floats and models in marine biogeochemistry"

_Biogeosciences, 2018_

## Referee Comment (RC1) · M. Ribera d'Alcala' (Referee) · 20 Aug 2018

Referee comment on "Merging bio-optical data from Biogeochemical-Argo floats and models in marine biogeochemistry" by Elena Terzić, Paolo Lazzari, Emanuele Organelli, Cosimo Solidoro, Stefano Salon, Fabrizio D'Ortenzio, and Pascal Conan

The paper discusses the results of the analysis of ∼1300 Biogeochemical ARGO profiles (Temperature, salinity, Chlorophyll fluorescence, downwelling irradiance at three wavelengths and downwelling PAR) generated in different regions of the Mediterranean sea, though covering a large portion of it, by 31 profilers in the years 2012-2016. The analysis is based on the comparison among measured profiles and profiles derived by merging different bio-optical models and 1D biogeochemical simulations based on a 3D

coupled biogeochemical model, the OGSTM-BFM (see text for refs). The wide scope motivation is that (P.2 L.22-24): Specific studies are required to demonstrate to what extent the assimilation of radiometric data can improve the model skill in simulating key biogeochemical variables (e.g. nutrients, primary productivity).

More specifically the authors want (P.2 L.32-34):

1) to show how it is possible to integrate BGC-Argo float bio-optical data and a simple 1-D model to investigate chlorophyll vertical dynamics; 2) [how] to use such a tool on a sufficiently large data set in order to test different bio-optical models

The text is unclear on a few key issues related to the protocol followed for the simulations (see below). Each simulated profile is generated using the vertical distributions of physical and chemical variables without considering horizontal processes, as the authors write on P.4 L.13-15 ..therefore implying that mass exchanges due to horizontal diffusion and baroclinic components of the (upper ocean) advection field are assumed to be smaller compared to vertical processes and biogeochemical dynamics The impact of this assumption depends on the time scale of integration and on what are the initial conditions of each run, which is not clearly explained.

A complementary scope is (P.5 L.13-14) ..[to assess] the possibility of using biogeochemical models also when [underwater] PAR measurements are not available, [comparing] the skill of different bio-optical models, which it is generally the rule.

The indicator for testing the performance of the models is the DCM depth, that obtained by the simulations vs. the observed depth, while a minor relevance is given to the DCM amplitude.

The main results of the study are: 1. an assessment of the performance of different formulations and/or parametrizations of the light penetration in the water column in relation to the concentrations of optically active components and 2. that PAR is more important than mixing and nutrients in determining the capability of the model in reproducing in situ chlorophyll profiles.

Indeed, testing different formulations and parametrizations in a model is useful not only to find the best performing model but, more importantly, to analyze the interplay among different mechanisms in generating observed pattern or dynamics. This part is often lacking in the discussion. For example the reason why different optical models produce different depths of the DCM varying with the area is not discussed.

More important, there is a key conceptual issue in the manuscript, at least from what I could grasp from its present version. The authors compare the chlorophyll vertical profiles, obtained from different bio-optical models and with different values of turbulent diffusivity, with those measured in situ, without discussing the impact on the profile of nutrients, phytoplankton loss due to grazing and all the other processes simulated by the OGSTM-BFM. I believe that the rationale for this is the assumption that the biogechemical module is always the same and then any differences in the results would depend only on the change of the specific driver tested. Even ignoring any possible non-linearity in specific processes, e.g., the nitrogen dependence of the photoacclimation by phytoplankton, the best performance of the model in reproducing the depth of the DCM cannot be attributed only to the tested drivers since equally important processes are in the background and not discussed at all, besides some mention to phosphate which is substantiated only by the model outputs. This makes me thinking that the authors consider the 'geochemical' fields produced by the OGSTM-BFM as real data instead than simulated data. This might be a reasonable assumption for large scale patterns but it is a little weaker for daily simulations in single sites that are moved in time. The effectiveness of a bio-optical model should be tested against IOPs or AOPs, as it is already been done also for BGC-ARGO proflers, not via an end product, i.e., chlorophyll a, whose concentration depend on many other processes. This would also help in clarifying which mechanisms drive the differences reported in Figs. 10 through 12.

In addition, it not clearly explained, or I might have missed where, if all the state variables simulated by the model were reset each day to the 3D model values for that day and that site, as one might guess from lines 30-33 on P.5 or if, as in a normal 1D simulations, they are produced by the model. In either case I guess some discrepancies should arise, which are neither mentioned nor discussed in the paper.

While acknowledging the effort invested in the study it looks a bit empirical and I am not convinced that it adds new knowledge to the existing one.

Besides solving a couple of issues mentioned in the detailed comments, I suggest to revise the paper analyzing in more detail what are the mechanisms driving the simulated differences and discussing in more detail the extent to which the OGSTM-BFM drives the DCM depth which is the prognostic variable that the author use to test the performance of the different sub-models tested.

Detailed comments

Abstract (It should be substantially re-written. Following are some suggestions).

L.3-4 ...Data set comprised of ..Argo Floats does not seem correct. I suggest to rephrase as: The present work is based on a dataset comprised of 1314 0-1000 m vertical profiles of biogeochemical and optical data measured by 31 Biogeochemical (BGC) Argo floats in the Mediterranean Sea from 2012 to 2016.

L.4 The data set was integrated in ...sounds a little confusing since the simulations are 1D. I suggest to rephrase as: 1-dimensional model simulations, using measured photosynthetically available radiation (PAR) profiles as light input, were then carried out for each profile along the trajectories of the floats.

L.6-7 The simulations were aimed to be consistent with data measured by float sensors, especially in terms of the deep chlorophyll maximum (DCM) depth. I suggest to rephrase as: The simulations were aimed at reproducing the profiles measured by float sensors, especially for what the deep chlorophyll maximum (DCM) depth concerns.

L.7-9 I suggest to rephrase as: We tested several light models to estimate their impact on modeled biogeochemical properties taking into account self-shading, derived from vertical chlorophyll distributions, and colored dissolved organic matter (CDOM) concentrations.

L.9-11 I suggest to rephrase as: The results, corroborated by the comparison with in-situ BGC-Argo profiles, illustrate how PAR penetration and vertical mixing modulate the dynamics of primary producers along the water column.

L.12 Highest?

L.13 Simulation results show also that...

L.14-15 After reading the paper I am not convinced that The approach here presented serves as a computationally smooth solution to analyse BGC-Argo floats data and to corroborate hypotheses on their spatio-temporal variability.

Intro

P.2 L.6 Density? More clear the high number of active BGC-Argos

P.2 L.7 ..numerical experiments of that kind. Unclear. Better: to analyze the predicting capability of bio-optical models, if this is the scope

P.2 L.19 ones

P.2 L.6-24 To better clarify the scope of the study it would be better to invert the sequence of the arguments. If the scope is to: ..to demonstrate to what extent the assimilation of radiometric data can improve the model skill in simulating key biogeochemical variables (e.g. nutrients, primary productivity) which comes as a possible improvement of what already done and sketched before, then this statement should come first. Then all the motivations for using Med data as a test case. If, alternatively, the scope is to improve our understanding of Med functioning then then all the paragraph should be changed accordingly. Reading the manuscript the first possibility seems to hold true.

Methods

P.3 L.17 ..were then vertically interpolated to a resolution of 1 m in the upper 400 m. Do the authors mean 'fitted'? If the sampling resolution was 1 m why to interpolate them? What about the data below 250 m? Were they extrapolated?

P.3 L.19-21 Could the authors be more explicit on which part of the Baird et al (2016) model they used and with which input variables? This can go in SI.

P.3 L.21 A second approach. There is no first before.

P.3 L.25 please rephrase as: ..measure Chl a concentration using as a proxy its fluorescence emission in the red band (690 nm) after blue excitation at 470 nm (Holm-Hansen et al., 1965)

P.3 l.27 remove it

P.5 L.20 ..levels

P.5 L.25 ..characterized regarding..? ..quantified using?

P.5 L.35 ..allow a gradual increase... decrease?

P.7 eq.1 I might be wrong but as written and with sigma-MLD = 0.3 the first term becomes negligible at the depth of 2 m

P.9 L.5-10 The whole paragraph is a little confusing because the authors introduce the seasonal mixing due to de-stratification without clarifying that this is likely taken into account by the measured change of the MLD and not by their formulation of mixing (Eq. 1).

P.10 L.29 remove as

Fig. 5 The legend could be compacted and the three figures could become one three multipanel figure

P.18 L.2 are hardly what? Constrained?

P.20 L.23-28 Do the authors implicitly assume that CDOM concentration is higher in

the WMed? This could said more explicitly.

P.27 L.12 The most fitting? May be: The best alternatives to fit the data.

---

## Author Comment (AC1) · 24 Aug 2018

We thank the reviewer for the detailed comments, which will be surely incorporated into a new revision. Below we list the major points raised by the reviewer with our propositions to fulfil the requirements (the comments of the reviewer are between quotes).

1 - "The text is unclear on a few key issues related to the protocol followed for the simulations including time scale of the simulation and initial conditions"

REPLY: we agree, and we are going to better specify the simulation protocols.

2 - "The indicator for testing the performance of the models is the DCM depth, that obtained by the simulations vs. the observed depth, while a minor relevance is given to the DCM amplitude."

[Figure]

REPLY: actually, in addition to the DCM depth we considered and showed also the correlation between simulated versus observed Hovmoeller diagrams for chlorophyll (point to point match), in this case the model skill results to be fairly good at R=0.75. Therefore, we considered not only the depth variability of DCM, but the whole signalÂă(DCM amplitude, thickness, temporal dynamic).Âă In any case, in the revised version we will further extend this part and make it more clear.

3 - "Indeed, testing different formulations and parametrizations in a model is useful not only to find the best performing model but, more importantly, to analyze the interplay among different mechanisms in generating observed pattern or dynamics. This part is often lacking in the discussion. For example, the reason why different optical models produce different depths of the DCM varying with the area is not discussed."

REPLY: we agree with the reviewer that the modelling approach here described can be useful to analyse the interplay between different mechanisms. We will expand this interesting part and further analyse the mechanism of the DCM gradients formation, including relevance of nutrients dynamics and grazing.

4 - "The effectiveness of a bio-optical model should be tested against IOPs or AOPs, as it has already been done also for BGC-ARGO profilers, not via an end product, i.e., chlorophyll a, whose concentration depend on many other processes. This would also help in clarifying which mechanisms drive the differences reported in Figs. 10 through 12."

REPLY: we agree with the reviewer and we are going to expand the manuscript, also accordingly to what planned for point 3.

5 - "While acknowledging the effort invested in the study it looks a bit empirical and I am not convinced that it adds new knowledge to the existing one."

REPLY: to the best of our knowledge, this is the first time that such a model configuration is presented. Moreover, the skill of the model is good, and our approach is

potentially applicable to other regions covered by BGC-Argo floats. Therefore, we believe that the present work adds new knowledge to the existing one. In particular, the good fit between observed data and the Reference simulation (forced by experimental BGC-Argo float data) support the underpinning assumptions related to: 1) relevance of vertical versus horizontal processes on spatial and temporal scales considered, 2) essential processes and kinetics coded in the biogeochemical model.

Additionally, we provide estimates on the impact of different bio-optical modules when data are not available.

We will surely improve the text of the manuscript to make these results more evident.

Minor comments will be also thoroughly addressed in the review.

Best Regards

---

## Referee Comment (RC2) · Anonymous Referee #2 · 27 Aug 2018

GENERAL COMMENTS

The authors used a number of vertical profiles from BIO ARGO floats (1314 profiles) in the Mediterranean and merged them with a one dimensional biogeochemical model. The aim of the study was to alter the optical component of the model and study the effect it has on model simulations, specifically on the chlorophyll profile. The authors also showed the effect vertical mixing has on the shape of the chlorophyll profiles. They have demonstrated that bio-optical data from the floats are useful not only for model data comparison, but also as forcing in the model, which in my take is the biggest plus of the work. I complement the authors on their effort combing the data with the model.

The work is well presented and concise. I think the manuscript is well suited to be published in this journal. My suggestion would be to expend some technical aspects,

which I outline in more detail with specific comments. These comments are aimed mostly to expand the information in the text.

SPECIFIC COMMENTS

P5 L30 How good is the matchup between the measured chlorophyll profiles and the modeled profiles taken for the initial conditions from the reanalysis?

P5 L22 If I am correct the governing equations for photosynthesis can be found in Lazzari et al. (2012) Appendix B and the remaining equations in Supplementary material of that paper? Please indicate this in more details.

P7 Perhaps writing a generic one dimensional equation for the vertical distribution of phytoplankton would be of some help to the non-expert readers of the paper. It would also help to elucidate the mathematical formulations of the various processes which are referred to later on in the text, such as mixing and light attenuation.

P17 Section 3.2 Some good references for this discussion are: Ryabov & Blasius (2014) The American Naturalist, Huisman et al. (2002) The American Naturalist, Huisman et al. (2004) Ecology, and one with a historical note: Ryabov & Blasius (2008) Mathematical Modelling of Natural Phenomena.

P7 L19 Does this imply that you have also averaged measured chlorophyll in the 15 m depth intervals along with calculated Kd and then pared them up in the regression? Please clarify.

P7 L24 Why are there brackets around ln(Ed)?

P9 Figure 2 The depth of the deep chlorophyll maximum is taken as a metric for the model and the model is proven to be very good at predicting the deep chlorophyll maximum depth. However, there are other measures beside this that can be used: surface chlorophyll concentration, chlorophyll concentration at the depth of the maximum and width of the profile. It would be interesting to see this comparison as a scatter plot.

P26 L8 Not quite sure if "irradiance propagation" is a correct term. Light propagates and irradiance is a measure of the light intensity per unit surface. Please change to "irradiance profile".

P26 L9 Change "position" to "depth".

TECHNICAL CORRECTIONS

I have noticed that in some places units are written with superscript (e.g. m s-1) and in some with a slash (e.g. m/s). Please opt for one to be consistent.

Also, in the figures chlorophyll concentration is written with small case letter c as "chl" and in the text it is written with capital letter C as "Chl". Again, please opt for one to be consistent. I would advise "Chl".

P6 Table 1 Wrong location of table caption. Should be above the table.

P6 Table 2 Wrong location of table caption. Should be above the table.

P3 L7 Units are in italics. Please change to upright.

P3 L10 Units are in italics. Please change to upright.

P7 L7 Mussing full stop at the end of the sentence.

P7 L16 Change "BCG-Argo" to "BGC-Argo".

P8 L22 Units are in italics. Please change to upright.

P10 L6 Units are in italics. Please change to upright.

P10 L18 Missing full stop after "sections".

P17 L9 Remove extra spacing before "where".

P26 L9 Change "what found" to "what was found" or "what has been found".

---

## Author Comment (AC2) · 28 Aug 2018

We thank very much Rev.2 for the positive and encouraging review. We will address the specific and technical comments and add the suggested references in order to expand the information provided in the submitted manuscript.
* * *

---

## Author Response (AR1)

Dear Editor,

we thank both reviewers for their useful comments. By following their suggestions, we will certainly be able to improve the quality of our manuscript. Below we attach a reply to all the points raised, where the reviewers' comments are in black, whereas our replies are indented in blue.

> To easily refer to the different reviews, we added a reference to each major point raised by the two Reviewers, labelled as "Rx.n", where "x" = n. of Reviewer, and "n" = n. of point.

**EDITOR COMMENTS**

The topic of your manuscript has been evaluated as of significant scientific interest and the overall presentation of good quality. However, both reviewers have raised some issues that I would urge you to constructively attend to, in particular, regarding the interpretation of DCM patterns produced by model simulations and how other processes (i.e. nutrient dynamics, grazing...) not discussed in the text affect patterns observed.

> We expanded the manuscripts as suggested by the Editor and Reviewers focusing on a better interpretation of the impact of nutrients and their interplay with different bio-optical models.

> This enabled a more complete and clearer overview of the patterns produced by the model compared to chlorophyll and radiometric data measured by BGC-Argo floats.

> We revised the introduction in order to make it more readable, as well as focused on the objectives and novelties of the present manuscript.

**REVIEWER #1**

Referee comment on "Merging bio-optical data from Biogeochemical-Argo floats and models in marine biogeochemistry" by Elena Terzić, Paolo Lazzari, Emanuele Organelli, Cosimo Solidoro, Stefano Salon, Fabrizio D'Ortenzio, and Pascal Conan

The paper discusses the results of the analysis of ~1300 Biogeochemical ARGO profiles (Temperature, salinity, Chlorophyll fluorescence, downwelling irradiance at three wavelengths and downwelling PAR) generated in different regions of the Mediterranean sea, though covering a large portion of it, by 31 profilers in the years 2012-2016. The analysis is based on the comparison among measured profiles and profiles derived by merging different bio-optical models and 1D biogeochemical simulations based on a 3D coupled biogeochemical model, the OGSTM-BFM (see text for refs). The wide scope motivation is that (P.2 L.22-24): Specific studies are required to demonstrate to what extent the assimilation of radiometric data can improve the model skill in simulating key biogeochemical variables (e.g. nutrients, primary productivity).

**R1.1**

More specifically the authors want (P.2 L.32-34):

1) to show how it is possible to integrate BGC-Argo float bio-optical data and a simple 1-D model to investigate chlorophyll vertical dynamics; 2) [how] to use such a tool on a sufficiently large data set in order to test different bio-optical models

The text is unclear on a few key issues related to the protocol followed for the simulations (see below). Each simulated profile is generated using the vertical distributions of physical and chemical variables without considering horizontal processes, as the authors write on P.4 L.13-15 ..therefore implying that mass exchanges due to horizontal diffusion and baroclinic components of the (upper ocean) advection field are assumed to be smaller compared to vertical processes and biogeochemical dynamics The impact of this assumption depends on the time scale of integration and on what are the initial conditions of each run, which is not clearly explained.

> The time scale of the simulations corresponds to the typical length of time-series provided by the BGC-Argo float during the period 2012-2016 (11 months on average).

> The initial conditions of each simulation, carried out by the OGSTM-BFM coupled physical-biogeochemical model, are provided by the outputs of the reanalyses of the MedBFM model system (composed by the OGSTM-BFM and the 3DVarBio assimilation scheme for surface chlorophyll from satellite, reanalyses released within the Copernicus Marine Environment Monitoring Services) at the corresponding spatial and temporal points of the float deployment. After the initialization, the model evolves without further assimilation of biogeochemical data from the 3D model, and it is not reinitialized. The simulation setup will be more extensively described in the revised version of our manuscript.

> We agree with the reviewer that the time scales are important. In particular, we hypothesize that in the experiments considered, several forcings like PAR and mixing are most important on short time scales, whilst other forcings (related to lateral advection of nutrients, for example) act on longer time scales by the modulation of subsurface nutrients inventories. However, we think that an extensive analysis of other mechanisms acting on the horizontal plane or along

isopycnal surfaces interacting with the float trajectory is beyond the scopes of our work. In any case, the analysis of the discrepancies between the 1D model results and the BGC-Argo float data can support the idea that when model and observations significantly disagree, physical and biogeochemical interactions not related to vertical processes may have a substantial role in the representation of the chlorophyll characteristics, not fully resolved by our 1D model framework.

**R1.2**

A complementary scope is (P.5 L.13-14) ..[to assess] the possibility of using biogeochemical models also when [underwater] PAR measurements are not available, [comparing] the skill of different bio-optical models, which it is generally the rule.

The indicator for testing the performance of the models is the DCM depth, that obtained by the simulations vs. the observed depth, while a minor relevance is given to the DCM amplitude.

The main results of the study are: 1. an assessment of the performance of different formulations and/or parametrizations of the light penetration in the water column in relation to the concentrations of optically active components and 2. that PAR is more important than mixing and nutrients in determining the capability of the model in reproducing in situ chlorophyll profiles.

Indeed, testing different formulations and parametrizations in a model is useful not only to find the best performing model but, more importantly, to analyze the interplay among different mechanisms in generating observed pattern or dynamics.

To better illustrate the effects of the parameterizations on the model indicators, we performed a number of experiments. In the first experiment we partitioned the BGC-Argo floats in couples: each couple is composed by one BGC-Argo float belonging to the western basin and one to the eastern basin, by random selection. Then, for each couple, we switched the initial conditions for nutrients, which allows to estimate the impact on DCM depth. The results are shown in the following scatter plots (Fig. R1):

[Figure]

**Fig. R1. Scatter plots of DCM depth derived for the REF simulation (left) and with the "East-West" switching technique described in the text (right).**

The plots evidence how inverting the initialization of the nutrients does not significantly alter the results in terms of DCM depth. We obtain a reduction of the slope from 0.81 to 0.62, thus it seems that the role of nutrients is secondary compared to light in DCM depth regulation.

Performing the same operation by switching light instead of nutrients is technically more difficult, however, we provided a second experiment to appraise the role of light (and other selected key parameters). This experiment consists in a sensitivity analysis following a similar technique as shown in Huisman et al. (2004). More specifically, we selected two BGC-Argo floats (*lovbio018c* for the east Med and *lovbio067c* for the west Med) and a couple of parameters [Phosphate, Light] and then performed 21x21 simulations (per each float) applying bivariate perturbations. This technique allows to better understand the driving mechanisms for DCM depth variability.

In the revised version of our manuscript, we used such analyses to evaluate the model sensitivity and to add some considerations to the results.

[Figure]

**Fig. R2. Sensitivity analysis of DCM depth perturbing *LIGHT* and initial conditions of *PO₄* [both by an uniform factor] along the water column. 'R' marks the reference values. The BGC-Argo float here reported is the *lovbio018c*. Each pixel is a full simulation, for a total of *21x21* simulations. The DCM depth is averaged over the simulation period.**

The plot reported in Fig.R2 shows how the DCM position is affected by light and nutrient ($PO_4$) perturbations. As shown in Fig.R2, perturbing $PO_4$ of 50% has a minor effect on DCM depth position, whilst perturbing light has a larger impact. Same results hold in the case of the other BGC-Argo float considered (*lovbio067c*). Such experiments were carried out also on additional indicators, such as DCM width and DCM values, as suggested by the other reviewer.

**R1.3**

This part is often lacking in the discussion. For example the reason why different optical models produce different depths of the DCM varying with the area is not discussed. More important, there is a key conceptual issue in the manuscript, at least from what I could grasp from its present version. The authors compare the chlorophyll vertical profiles, obtained from different bio-optical models and with different values of turbulent diffusivity, with those measured in situ, without discussing the impact on the profile of nutrients, phytoplankton loss due to grazing and all the other processes simulated by the OGSTM-BFM.

We focused on discussing the DCM depth because it is the indicator measured by the BGC-Argo floats that we principally considered in this manuscript. We do not have available synoptic data measured by the BGC-Argo floats (e.g. nutrient concentrations) to corroborate the other outputs produced by the model. Therefore, for variables different from chlorophyll, we can provide at most an evaluation of the impact the bio-optical models have on them. The additional impacts on nutrients and phytoplankton grazing are also driven by the same changes in the parameterizations selected. We assume that on the time scales of the simulation [11 months], the most important mechanisms (light, mixing) are included in the model, thus, the variability of simulated profiles of nutrients should be realistic. As an additional analysis, we perturbed initial conditions of nutrients to evaluate the effect of DCM properties. This allows to explore mechanisms controlling chlorophyll dynamics. Additionally, we perturbed initial condition only for the BGC-Argo floats deployed in the western basin to evaluate the impact on reproducing gradients.

**R1.4**

I believe that the rationale for this is the assumption that the biogeochemical module is always the same and then any differences in the results would depend only on the change of the specific driver tested. Even ignoring any possible non-linearity in specific processes, e.g., the nitrogen dependence of the photoacclimation by phytoplankton, the best performance of the model in reproducing the depth of the DCM cannot be attributed only to the tested drivers since equally important processes are in the background and not discussed at all, besides some mention to phosphate which is substantiated only by the model outputs. This makes me thinking that the authors consider the 'geochemical' fields produced by the OGSTM-BFM as real data instead than simulated data. This might be a reasonable assumption for large scale patterns but it is a little weaker for daily simulations in single sites that are moved in time. The effectiveness of a bio-optical model should be tested against IOPs or AOPs, as it is already been

done also for BGC-ARGO profilers, not via an end product, i.e., chlorophyll a, whose concentration depend on many other processes. This would also help in clarifying which mechanisms drive the differences reported in Figs. 10 through 12.

> We agree with the reviewer and, as suggested, we propose to compare the skill of the models using directly the AOPs, in particular the average irradiance attenuation and the maximum attenuation along the vertical compared to REF that adopts measured values. Considering also the role in nutrients affecting attenuation. Nonetheless, we think that it is important to keep also the comparison with the DCM depth because it is the end product we are mainly interested in the present manuscript.

**R1.5**

In addition, it not clearly explained, or I might have missed where, if all the state variables simulated by the model were reset each day to the 3D model values for that day and that site, as one might guess from lines 30-33 on P.5 or if, as in a normal 1D simulations, they are produced by the model. In either case I guess some discrepancies should arise, which are neither mentioned nor discussed in the paper.

> As explained before, the 1-D model is initialized with the 3-D model only at the first step of the simulation. We mention the fact that neglecting lateral inputs could produce effects that the present methodology cannot replicate. We do not want to stress the dependence on 3-D model configuration because a possible generic application carried out with this approach, e.g. based on the global ocean BGC-Argo float dataset, could be possibly performed independently from any 3D model, and the initialization for nutrients might be based on data available from climatology repositories.

**R1.6**

While acknowledging the effort invested in the study it looks a bit empirical and I am not convinced that it adds new knowledge to the existing one. Besides solving a couple of issues mentioned in the detailed comments, I suggest to revise the paper analyzing in more detail what are the mechanisms driving the simulated differences and discussing in more detail the extent to which the OGSTM-BFM drives the DCM depth which is the prognostic variable that the author use to test the performance of the different sub-models tested.

> We agree to further expand this part in order to make more evident the differences between the REF simulation and the alternative models in terms of skill and formulation. We will add additional indicators in the analysis of REF results (as mentioned in point R2.7).

Detailed comments

Abstract (It should be substantially re-written. Following are some suggestions).

L.3-4 ...Data set comprised of ..Argo Floats does not seem correct. I suggest to rephrase as: The present work is based on a dataset comprised of 1314 0-1000 m vertical profiles of biogeochemical and optical data measured by 31 Biogeochemical (BGC) Argo floats in the Mediterranean Sea from 2012 to 2016. L.4 The data set was integrated in ...sounds a little confusing since the simulations are 1D. I suggest to rephrase as: 1-dimensional model simulations, using measured photosynthetically available radiation (PAR) profiles as light input, were then carried out for each profile along the trajectories of the floats. L.6-7 The simulations were aimed to be consistent with data measured by float sensors, especially in terms of the deep chlorophyll maximum (DCM) depth. I suggest to rephrase as: The simulations were aimed at reproducing the profiles measured by float sensors, especially for what the deep chlorophyll maximum (DCM) depth concerns. L.7-9 I suggest to rephrase as: We tested several light models to estimate their impact on modeled biogeochemical properties taking into account self-shading, derived from vertical chlorophyll distributions, and colored dissolved organic matter (CDOM) concentrations. L.9-11 I suggest to rephrase as: The results, corroborated by the comparison with in- situ BGC-Argo profiles, illustrate how PAR penetration and vertical mixing modulate the dynamics of primary producers along the water column. L.12 Highest? L.13 Simulation results show also that... L.14-15 After reading the paper I am not convinced that The approach here presented serves as a computationally smooth solution to analyse BGC-Argo floats data and to corroborate hypotheses on their spatio-temporal variability.

> We agreed to follow the reviewer's suggestions and modified the abstract accordingly

Intro
P.2 L.6 Density? More clear the high number of active BGC-Argos

> We reformulated this part.

P.2 L.7 ..numerical experiments of that kind. Unclear. Better: to analyze the predicting capability of bio-optical models, if this is the scope

> Ok

P.2 L.19 ones

> Ok

P.2 L.6-24 To better clarify the scope of the study it would be better to invert the se- quence of the arguments. If the scope is to: ..to demonstrate to what extent the assimi- lation of radiometric data can improve the model skill in simulating key biogeochemical

variables (e.g. nutrients, primary productivity) which comes as a possible improvement of what already done and sketched before, then this statement should come first. Then all the motivations for using Med data as a test case. If, alternatively, the scope is to improve our understanding of Med functioning then then all the paragraph should be changed accordingly. Reading the manuscript the first possibility seems to hold true.

> We reformulated our introduction to make it clearer.

Methods

P.3 L.17 ..were then vertically interpolated to a resolution of 1 m in the upper 400 m. Do the authors mean 'fitted'? If the sampling resolution was 1 m why to interpolate them? What about the data below 250 m? Were they extrapolated?

> Yes the data were fitted in order to be regularly spaced as in the case of the model, the data below 250 were extrapolated.

P.3 L.19-21 Could the authors be more explicit on which part of the Baird et al (2016) model they used and with which input variables? This can go in SI.

> We will add more details concerning the correction applied to PAR in the supplementary material

P.3 L.21 A second approach. There is no first before.

> The corrections we mention are related to the corrections of PAR from planar to scalar, the first is with the formula by Baird et al. (2016), the second is by means of constant correction, we will clarify the text accordingly. For the sake of clarity we left in the text only the selected approach, Baird et al. (2016).

P.3 L.25 please rephrase as: ..measure Chl a concentration using as a proxy its fluorescence emission in the red band (690 nm) after blue excitation at 470 nm (Holm-Hansen et al., 1965)

> We changed the text and omitted this part in the updated version.

P.3 l.27 remove it

> Ok

P.5 L.20 ..levels

> Ok

P.5 L.25 ..characterized regarding..? ..quantified using?

> Ok

P.5 L.35 ..allow a gradual increase. . . decrease?

> Ok, we can rephrase putting "allowing a vertically smooth transition between mixed and stratified layers."

P.7 eq.1 I might be wrong but as written and with sigma-MLD = 0.3 the first term becomes negligible at the depth of 2 m

> The denominator in the argument of the exponential is sigma * MLD, in this way the MLD depth modulates the shape of the mixing profiles in terms of variance of the Gaussian.

P.9 L.5-10 The whole paragraph is a little confusing because the authors introduce the seasonal mixing due to destratification without clarifying that this is likely taken into account by the measured change of the MLD and not by their formulation of mixing (Eq. 1).

> The explanation of the formula in Eq.1 should now make the paragraph more clear, in fact, the variance of the Gaussian profile depends on the MLD.

P.10 L.29 remove as

> Ok

Fig. 5 The legend could be compacted and the three figures could become one three multipanel figure

> Ok, we will try to compact the figure, our only concern is related to readability of the panels if they become too small.

P.18 L.2 are hardly what? Constrained?

> Hardly ever in a steady state condition. We mean that the system has time dependent forcing that prevent it to reach the steady state.

P.20 L.23-28 Do the authors implicitly assume that CDOM concentration is higher in the WMed? This could said more explicitly.

> Yes, this assumption derives from the preliminary analysis we carried out on Ed 380 BGC-Argo float profiles, for which the Kd 380 were derived. Since the diffusion attenuation coefficient as an apparent optical property depends more on the composition of the examined water body rather than the external light field (i.e. on IOPs), higher Kd values at that wavelength suggest a higher absorption, be it from CDOM and / or non-algal particles (NAP). From the same data set, after an extended analysis, it can be confirmed that a gradient in absorption between west and east Mediterranean is present.

P.27 L.12 The most fitting? May be: The best alternatives to fit the data.

Ok

REVIEWER #2

GENERAL COMMENTS

The authors used a number of vertical profiles from BIO ARGO floats (1314 profiles) in the Mediterranean and merged them with a one dimensional biogeochemical model. The aim of the study was to alter the optical component of the model and study the effect it has on model simulations, specifically on the chlorophyll profile. The authors also showed the effect vertical mixing has on the shape of the chlorophyll profiles. They have demonstrated that bio-optical data from the floats are useful not only for model data comparison, but also as forcing in the model, which in my take is the biggest plus of the work. I complement the authors on their effort combining the data with the model.

The work is well presented and concise. I think the manuscript is well suited to be published in this journal. My suggestion would be to expend some technical aspects, which I outline in more detail with specific comments. These comments are aimed mostly to expand the information in the text.

We thank the reviewer for the encouraging comments, below we reply to the points raised.

SPECIFIC COMMENTS

R2.1

P5 L30 How good is the matchup between the measured chlorophyll profiles and the modeled profiles taken for the initial conditions from the reanalysis?

The following scatter plot (Fig.R3) is equivalent to the one used for the REF model validation but restricted to the initial values taken respectively from reanalyses and the BGC-Argo float data. The number of samples is lower than in the case of the BGC-Argo float results, and therefore difficult to compare with the other scatter plot (Fig.R1). However, we may point out that model tends to overestimate the DCM position (Bias is approximately 7% of the mean DCM depth and the slope is 0.53).

[Figure]

Fig. R3. Same as Fig. R1 (left panel) but only for initial conditions.

R2.2

P5 L22 If I am correct the governing equations for photosynthesis can be found in Lazzari et al. (2012) Appendix B and the remaining equations in Supplementary material of that paper? Please indicate this in more details.

Ok, we propose to add a unambiguous reference: the equations are best summarized in the BFM manual where all the options including the ones used in the present simulation are reported. The supplementary material included in this manuscript contains the biogeochemical parameters that activate the correct options used in the present simulations.

R2.3

P7 Perhaps writing a generic one dimensional equation for the vertical distribution of phytoplankton would be of some help to the non-expert readers of the paper. It would also help to elucidate the mathematical formulations of the various processes which are referred to later on in the text, such as mixing and light attenuation.

We agree with the reviewer. We plan to add the general mathematical equation applied to each tracer:

$$\partial_t C_i(z,t) = \partial_z[D_v(z,t)\partial_z C_i(z,t)] + v_{sink,i}\partial_z C_i(z,t) + BFM_i(T,S,PAR,\overline{C}(z,t))$$

where $Ci$ are the biogeochemical tracers simulated (i=1,50), $Dv$ is the vertical eddy diffusivity derived from Eq.1 [reported in the first submission of the manuscript]. $v_{sink}$ is the sinking velocity, $BFMi$ is the reaction term due to biogeochemical processes for the tracer $Ci$. T, S, PAR are data measured by the BGC-Argo float.

**R2.4**

P17 Secti 3.2 Some good references for this discussion are: Ryabov & Blasius (2014) The American Naturalist, Huisman et al. (2002) The American Naturalist, Huisman et al. (2004) Ecology, and one with a historical note: Ryabov & Blasius (2008) Mathematical Modelling of Natural Phenomena.

Thank you for the recommendations. We found the literature very helpful and have in turn added these very useful references, in particular to comment the theoretical aspects of the simulated profiles.

**R2.5**

P7 L19 Does this imply that you have also averaged measured chlorophyll in the 15 m depth intervals along with calculated Kd and then pared them up in the regression? Please clarify.

Yes, we proceeded exactly in this way, which has been specifiedin the revised manuscript.

**R2.6**

P7 L24 Why are there brackets around ln(Ed)?

The brackets [] are a typo, we will correct it.

**R2.7**

P9 Figure 2 The depth of the deep chlorophyll maximum is taken as a metric for the model and the model is proven to be very good at predicting the deep chlorophyll maximum depth. However, there are other measures beside this that can be used: surface chlorophyll concentration, chlorophyll concentration at the depth of the maximum and width of the profile. It would be interesting to see this comparison as a scatter plot.

Our initial idea to focus mainly on the shape of the profile was dictated by the complexity of the transformation of fluorescence profiles to chlorophyll concentration values. For this reason, we thought that comparing the simulated DCM depth versus measurements was the most robust action to take. We already included an evaluation of the surface concentration for the stratified period to compare the effect of constant versus diel variation in PAR.

Following the reviewer's suggestions, we show also the DCM width and the DCM magnitude. The DCM width is operationally defined by means of a Gaussian fit and the thickness is computed in the range +/- sigma/2 from the maximum.

[Figure]

[Figure]

**Fig. R4. Scatter plot of DCM thickness as defined in the text. Left panel reports REF simulation ($D_v^{background}=10^{-4}$ m$^2$s$^{-1}$), right panel shows MLD04 simulation ($D_v^{background}=10^{-6}$ m$^2$s$^{-1}$). The thickness is defined as +/- sigma/2 computed on the vertical profiles by means of a Gaussian fit.**

As shown in Fig. R4, the correlation between modeled and measured DCM thickness is lower compared to the DCM depth statistics. The model has a minimum thickness of approximately 15 meters, whereas data reach in some cases 5 meters. As explained in the first version of the paper, background diffusivity regulates the shape of relative maxima. Spatial variability of the background diffusivity coefficient ($D_v^{background}$) in the Mediterranean Sea could be responsible for the higher variability in the DCM thickness observed in data versus model. In the experiments considered as alternative MLD models (MLD01, MLD02, MLD03, MLD04), we changed the $D_v^{background}$ parameter for all BGC-Argo floats for the same amount. The comparison between REF and MLD04 with extreme values of $D_v^{background}$ evidences how, on average, the DCM thickness reduces as diffusivity reduces (Fig. R4, right).

The case of chlorophyll concentration at DCM is more complex. Measured chlorophyll concentration fluctuates in the DCM, and an investigation of the possible underlying mechanisms (e.g. presence of Rossby or Kelvin waves, or other non-linear effects) go beyond the scope of the present paper.

We show here the median chlorophyll in the DCM productive layer (+/- sigma/2) for each BGC-Argo float (Fig.R5). In general, simulations tend to underestimate chlorophyll concentration compared to BGC-Argo floats in the western Mediterranean. In the first version of the manuscript we emphasized how nutrients control the biomass in the DCM productive layer. We evaluated the effects of perturbing nutrients for the BGC-Argo floats deployed in the West Mediterranean by increasing the $PO_4$ concentration by a factor 2. The results are reported in Fig. R5.

[Figure]

**Fig. R5. Scatter plot of DCM chlorophyll concentration as defined in the text: median concentration of the REF (blue dots) and from the simulation increasing PO₄ (orange dots).**

The interesting result is that the skill in reproducing the DCM depth, Fig. R1(left), is almost the same between REF and REF with higher $PO_4$ (image not shown) so it could be possible to finely tune the initial conditions to maximize both the skills in terms of DCM value and DCM depth. But considering the fact that the measurements of concentration of chlorophyll as derived from fluorescence present some uncertainties, we prefer to keep the initialization as based on reanalysis.

For a more detailed overview of the quality control procedure for fluorescence profiles, see Organelli et al. 2017 (**https://www.earth-syst-sci-data.net/9/861/2017/** ) as a reference. We will underline this also in the following version of the manuscript.

**R2.8**

P26 L8 Not quite sure if "irradiance propagation" is a correct term. Light propagates and irradiance is a measure of the light intensity per unit surface. Please change to "irradiance profile".

Ok, we agree to substitute "irradiance propagation" with "irradiance profile".

**R2.9**

P26 L9 Change "position" to "depth".

Ok.

I have noticed that in some places units are written with superscript (e.g. m s-1) and in some with a slash (e.g. m/s). Please opt for one to be consistent.

> Ok.

Also, in the figures chlorophyll concentration is written with small case letter c as "chl" and in the text it is written with capital letter C as "Chl". Again, please opt for one to be consistent. I would advise "Chl".

> Ok, we will standardize the notation with Chl.

P6 Table 1 Wrong location of table caption. Should be above the table. P6 Table 2 Wrong location of table caption. Should be above the table. P3 L7 Units are in italics. Please change to upright.P3 L10 Units are in italics. Please change to upright. P7 L7 Mussing full stop at the end of the sentence. P7 L16 Change "BCG-Argo" to "BGC-Argo".P8 L22 Units are in italics. Please change to upright. P10 L6 Units are in italics. Please change to upright. P10 L18 Missing full stop after "sections". P17 L9 Remove extra spacing before "where".P26 L9 Change "what found" to "what was found" or "what has been found".

> Ok, we will apply the corrections listed above.

 List of relevant changes in the manuscript:

1. **Abstract** was rewritten
2. **Introduction** was rewritten
3. **Mehods** were revised
4. **Results and discussion** was substantially extended to include a more detailed analysis on the impact of nutrients on model results. To this end the number of simulations performed has been increased significantly. Additionally, comparison of optical model skill not only in terms of chlorophyll but also in term of Kd was included
5.  **Conclusions** were expanded including new results and insight derived from the additional simulation performed
6. **Supplementary material** was expanded

In the following part of the document we attach a comparison of the first submitted manuscript with the presently submitted one (R1) processed with latex diff software in order to show the changes performed.

[revised manuscript text omitted]

---

## Author Response (AR2)

Dear editor,

we considered all the proposed suggestions for the manuscript's correction. Below we reply to specific questions and uncertainties arising from the previous version of the text. We decided to remove some plots (Figs. 12, 13 and 15 from the first revision's version) without affecting the message of the manuscript and at the same time reducing its' length. The plot related to sensitivity analysis were moved in the Supplementary material.

**Pg.2, c1: There seems to be a bit of a contradiction here: if radiometric data is more accurate, how does it help to use it to estimate chlorophyll values that are then validated using the supposedly "inaccurate" BGC-Argo fluorescence-derived Chlorophyll?**
Measured Chl data could be inaccurate in terms of magnitude, but not in terms of DCM depth, even though it's derived from fluorescence. The validation is therefore more concentrated on the DCM depth rather than magnitude due to the uncertainties associated with the latter. We would keep the sentence as it is.

**Pg.2, c4: Why is there is no mention of the DCM. Isn't that your most important validation parameter?**
We agree with the editor and modified the paragraph accordingly.

**Pg.5, c1: I am not sure I understand the meaning of this sentence.**
It's indeed redundant within the scope of the present study, so we cancelled it.

**Pg.6, c1: Would it be possible to include the depth range or intervals from which the parametrizations are derived (or give the info in Table 2).**
$z_{max}$ in Table 2 is already the depth range. The regression analysis is carried out for mean values of depth layers of 15 m thickness, which has been specified also in Sect. 2.2.2.
We modified the Table 2 caption accordingly.

**Pg.6, c4: Shouldn't the units be 1m/s?**
Eddy diffusivity coefficients are expressed in $m^2/s$.

**Pg.9, c1: This section title is somewhat misleading. This section covers all of 22 pages of the manuscript and does not only describe the reference simulations but also sensitivity analysis of the REF model which is not announced in the introduction and methods.**
According to the editor's suggestion, we moved the paragraph related to the sensitivity analysis to the supplementary material.

**Pg.9, c2: What do the legend (names and colors) represent? I assume data from individual floats? over what time range each?**

Each color represents one BGC-Argo float. The chart reporting the WMO code of each BGC-Argo float is in the Supplementary material. We updated it by adding the float tracking period information.

**What is the p-value for the regression? (analysis of variance or t-test?) are the correlations statistically significant?**

t-test. Yes, the correlations are statistically significant. p-value < 0.005 (as reported in the caption of Fig.2)

**Pg.10, c3: Do the authors mean at the surface of the ocean, or do they mean something like "primarily" or "overall"?**

At the surface of the ocean, we modified the sentence accordingly.

**Pg.10, c5: Can you please explain what the "initial conditions statistics" mean?**

We evaluated the initial conditions based on reanalysis (used in the 3-dimensional model configuration) versus BGC-Argo float (Chl) data. This was additionally tested to make sure that the initial conditions of our 1-dimensional model do not degrade when integrated in time. Since it's not essential information we deleted this sentence.

**Further here R is in capital letters but not in the figure 2 for example.**

We corrected this as well.

**Pg.15, c2: I am not sure I understand the link between lateral advection and strong vertical gradients in nutrient inventories. Can this be clarified?**

Both lateral advection and vertical mixing could impact the nutrient inventory variability, but here we verify that the most important process is vertical mixing (regarding DCM depth features). As specified in the text, data driven mixing and vertical turbulence effects allow to simulate correctly the seasonal variability of the DCM depth.

We modified the paragraph, hopefully making it easier to understand.

**Pg.15, c9: What is the p-value for the regression? (analysis of variance or t-test?) are the correlations statistically significant?**

See reply for Pg.9, c2.

**Pg.15, c11: It would be helpful if the same units were used in all graphs (see figs. 3-6 where PAR is given as µmol/m2/s.**

The units are now uniform in all plots, i.e. we chose **µmol/m2/s**

**Pg.17, c1: This whole section could go in an appendix as these are additional experiments not mentioned in the introduction. See also previous comments.**

We followed the suggestion and moved the section in the Supplementary material.

**Pg.18, c1: Why is the CL1 simulation shown here. Shouldn't this be shown in the next sections?**

See reply above (pg.17, c1)

**What is the p value for the regression? (analysis of variance or t-test?) are the correlations statistically significant?**
See reply for Pg.9, c2.

**Please explain how average Chl 0-25m was estimated as well as what dataset (which BGC-Argo float) was used here.**
We calculated the mean value of Chl in the first 25 meters for all profiles when DCM was present. The same criterion was applied for both data and model.

**Pg.18, c4: Thickness? Biomass? Please specify what magnitude means.**
Biomass. We decided to remove this picture as mentioned in the beginning of the reply.

**Pg.19, c8: "indicate that surface"**
We don't refer only to surface nutrients.

**Pg.21, c1: Please explain how BIO and PO4 were estimated. Are those model results or field measurements?**
The sentence was modified in order to make it clear that we are talking about model results and not field measurements. There are no data available for phosphates from the BGC-Argo network.

**Pg.22, c1: Why not present the full results from these simulations here? Do the values in text correspond to results shown in Fig. 14?**
We prefer to keep only one summarizing plot to avoid having too many of them.

**Pg.23, c1: Why is RMSD negative? Ed. 6 given in p.8 should give positive values only.**
Yes we agree, we modified the figure by plotting only the positive axis.

**Pg.24, c3: I fail to see that fig. 16 (16a in particular) shows a east-west gradient since only residuals are plotted.**
We added another subplot with monthly climatology DCM for west and east separately. This should show the gradient described in the paper.

**Pg.25, c1: The statement based on which data/simulation?**
The statement is based on a reference we added also at the end of the sentence (Crispi et al., 2001).

**Pg.29, c1: I have noticed differences between the REF and measured profiles in this figure compared to Fig. 15. Can this be explained? are those data for different time periods? For the sake of comparison wouldn't it make sense to choose the same time period?**
They are the same, we checked again (both the red – REF - and dark blue – DATA - lines). The other curves are different because we consider different subsets of simulations.

**Pg.30, c6: The meaning of this sentence is unclear.**
We deleted it the second part of the sentence, which made it unclear.

**Referee comment** on the revised version

**I acknowledge the effort made by the authors to comply with referees' suggestions in the revised version but I think that the paper needs a further effort to be ready for being published. This for the following reasons:**

**1. One of the declared scopes of the paper (e.g., Title, p.1 l.5, p1 l.23-p.2 l.7) was to explore the advantage of assimilating Argo profiles in a coupled model. Reading the manuscript I understood that the assimilated variables were, in turn, PAR and chlorophyll a profiles to compute PAR with several bio-optical models. As for the mixed layer depth, which is one term to modulate the diffusivity profile, it is not clear if it responds to external forcing, e.g., p.5 l.5, or is also an assimilated variable. I might have missed this. If not, this should be clarified.**

As explained in Sect. 2.2.1, the mixed layer depth is computed from temperature and salinity measured from BGC-Argo floats and it's used in the mixing model.

**2. On the other hand, a significant part of the text is devoted to discuss not too much the feasibility of ARGO data assimilation in a coupled model but the mechanisms determining the DCM dynamics. This is interesting but, apparently, the authors do not analyze the basic mechanism behind the functioning of the DCM. At a first order of approximation the DCM is the depth where the upward diffusive nutrient flux is fully uptaken by, prevalently, phytoplankton. This is why the isolume is a good, first order, proxy for DCM depth. This has been discussed by Letelier e al. (L&O *49*(2), 508-519, 2004) and, more recently by Cullen (AnnRevMarSci, 2015) none of whom is cited in the paper. Of course, there might be phylogenetic or ontogenetic adaptations, but I assume that the model has a constant physiology for phytoplankton. I would hypothesize that increasing vertical diffusion should certainly increase the carrying capacity, and therefore, the DCM amplitude, which is what the authors observe, but it should also move the DCM depth upwards to reach a new steady state where the nutrient flux is utilized at the higher rate because of higher photon flux. Diffusion does also disperse cells but the authors focus mostly on this aspect, i.e., the thickness of the DCM not on its depth dependence on diffusion and vertical gradients.**

Yes, we evaluated this in a sensitivity analysis, analogous to the one reported in the 20x20 bivariate perturbations experiment (see Supplementary material). Perturbing mixing for 4 orders of magnitude ($10^{-6}$ to $10^{-2}$ m$^2$/s), the average difference is around 10 meters in DCM depth, with higher DCM when mixing is higher, as mentioned above by the reviewer. We added suggested references with a brief discussion on the role of mixing.

**3. The authors focus on phosphate as the possible driving nutrient. It may depend on the existing paradigm that phosphate is the 'limiting' nutrient in the Mediterranean Sea. It might be interesting to examine the nitrate behavior. However, the intriguing**

**pattern is that the phosphate concentration in the WMED is approximately double than in the EMED at the same isolume but the chlorophyll is more or less the same, for what can be seen from figures 3 to 6. How the authors interpret this, since the phytoplankton physiology should be the same? May be that the similarity is a bias of the graphic representation.**

We agree with the reviewer's comments. In future we plan to consider also nitrate data from BGC-Argo floats, which could be an additional validation parameter.

The Chl content at DCM appears higher in the western basin, but the response is nonlinear: double nutrient concentration does not directly imply double Chl concentration.

In the sensitivity analysis section (Supplementary material) we evaluated the response in Chl concentration by further increasing the concentration of nutrients. The effect of increasing nutrients is also evaluated in terms of self-shading in the section related to bio-optical models.

**4. Linked to the above is the sensitivity of the DCM depth to phosphate (nutrient?) profile. Swapping East and West Argo profilers the authors (see response) state that there is no significant effect. Indeed, the slope of the model DCM vs observed DCM depth shows that the model underestimates the DCM depth for deep DCM and slightly overestimates the depth for shallow DCM, a pattern that is not discussed. More important, when they swap the profiles the model enhances this feature, which I would interpret as the fact that the higher irradiance in the EMED produce shallower DCM than in the real environment and the opposite occurs in the WMED. The scatter plot in Figure 2 does not allow a simple geo-localization of the Argos but, in any case, I would not consider the result of the analysis as a demonstration that nutrient profile has a minor role in determining the DCM depth.**

As explained in the conclusions section, we are not stating that the role of nutrients in shaping the DCM is absent. The evaluation of the role of nutrients in this manuscript is performed in two stages:

1) In the REF simulation only direct effects of nutrients are accounted for, e.g. in relation to vertical diffusion and the corresponding nutrient upward fluxes. In this case we observed that light appears to play a mayor role in shaping the gradient.

2) If we consider indirect effects of nutrients on light propagation, we see how nutrients play a role trough self-shading. This is demonstrated through the analyses which used alternative bio-optical models that account for self-shading effects.

**5. More important, even not being an English mother language, I think that the text should be revised both in the wording and in the way the work done is presented. I still found some parts hard to follow and to connect to the others.**

We followed the reviewer's suggestion and revised the text thoroughly.

In the following part of the document we attach a comparison of the first submitted manuscript with the presently submitted one (R2) processed with latex diff software in order to show the changes performed.

[revised manuscript text omitted]